# Investigation and Research of High-Performance RF MEMS Switches for Use in the 5G RF Front-End Modules

**DOI:** 10.3390/mi14020477

**Published:** 2023-02-18

**Authors:** Alexey Tkachenko, Igor Lysenko, Andrey Kovalev

**Affiliations:** Design Center of the Microelectronic Component Base for Artificial Intelligence Systems, Southern Federal University, Taganrog 347922, Russia

**Keywords:** microelectromechanical systems, MEMS, radio-frequency, RF, RF MEMS, switch, methodology of designing, high-performance, 5G mobile networks systems

## Abstract

In this article, based on the developed methodology, the stages of designing two designs of high-performance radio-frequency single-pole single-throw microelectromechanical switches are investigated. These radio-frequency microelectromechanical switches are designed to operate at a central resonant frequency of 3.6 GHz and 3.4 GHz, respectively, as well as to work both in mobile communication devices and in the design of the architecture of 5G mobile networks, in particular in arrays of integrated antennas and radio-frequency interface modules. The manufacture and study of two designed structures are researched. For the first manufactured experimental sample in the open state the insertion loss is no more than −0.69 dB and the reflection loss is −28.35 dB, and in the closed state the isolation value is at least −54.77 dB at a central resonant frequency of 3.6 GHz. For the second in the open state the value of the insertion loss is no more than −0.67 dB and the reflection loss is −20.7 dB, and in the closed state the isolation value is not less than −52.13 dB at the central resonant frequency of 3.4 GHz. Both manufactured experimental samples are characterized by high linearity, as well as a small value of contact resistance in the closed state.

## 1. Introduction

In the last decade, devices manufactured using microelectromechanical systems (MEMS) technology have undergone tremendous development in various fields of information and communication technologies. In particular, recent trends in the interaction of integrated circuit (IC) components with the external environment have prompted the development of innovative devices and technologies that can combine electrical properties with several different physical areas. In particular, MEMS devices use a miniature movable structure, the movement or position of which can be controlled by means of an electrostatic, thermal, magnetic, liquid or electromagnetic (EM) activation mechanism. Reliability, compactness and a high degree of integration were among the key factors for achieving successful products in the technology market, such as ST Microelectronics accelerometers for car airbags [1], Nintendo Wii controllers [2], Apple iPhone gyroscopes [3], inkjet devices [4] and Texas Instruments digital micro-mirrors for projectors [5].

One of the most promising areas of application of MEMS technology is associated with circuits and devices of ultra-high frequencies (UHF) and radio-frequency (RF) communication. In recent years, the introduction and dissemination of new wireless communication standards, in particular the fifth generation of mobile radio communications—5G, has set new challenges in the development of hardware for transmitter, receiver and transceiver devices [6,7,8,9]. At the level of one individual electronic device, the most important characteristics to consider are: low power consumption, high linearity and high bandwidth (reconfigurability/flexibility), which should characterize each component to ensure high performance of wireless radio communication systems and 5G mobile networks [10].

5G will implement a completely different paradigm compared to 4G and 4G-LTE. Some of the services we use today, such as Wi-Fi internet access and video streaming, will be covered by 5G coverage along with classic features such as voice calls and mobile internet access. It is also important to note that machine-to-machine (M2M) communication data is expected to be transmitted over 5G protocols. Examples of M2M applications are autonomous vehicles, remote surgery, remote manufacturing and smart cities. In other words, a significant portion of the Internet of Things (IoT) and Internet of Everything (IoE) data traffic will depend on 5G networks. Obviously, the data throughput requirement is going to be huge. Many forecasts call for a 1000-fold increase in 5G transmission capacity over 4G-LTE, providing 10 Gb/s for each individual user. In addition, the data transfer delay will need to be drastically reduced to a millisecond level. To understand the importance of the latter requirement, one can simply wonder how low latency can be critical for applications such as vehicle-to-vehicle (V2V) communication. Finally, more importantly, using M2M applications, cloud computing, IoT, IoE and so on will require a greater symmetry between the downlink and uplink bandwidth of the 5G standard [10].

Since this new technology is implemented and used by a wide class of components, such as broadband switches, switching blocks, multi-position impedance tuners and multi-position phase shifters, reconfigurable filters, programmable step attenuators and hybrid devices, as well as miniature antennas and arrays, these components will require self-updating to support the excellent performance of 5G networks. Key RF subsystems in 5G RF transceivers include antennas, tunable filters, power amplifiers and multiple-input multiple-output antennas (MIMOs) [10,11,12].

At the same time, it can be noted that in all 5G RF systems or subsystems (mobile Wi-Fi solutions, envelope tracking solutions, antenna and impedance reconfiguration solutions, integrated/discrete solutions, spaced receiving modules, RF front-end modules (RFFEs)), from a hardware point of view, the RF switch is one of the most fundamental and important components that is used to route signals along RF signal transmission paths with a high degree of efficiency [13,14,15,16,17]. Its RF characteristics, switching time, RF signal power value and reliability can directly affect the corresponding properties and performance of 5G applications. At the same time, in concordance with the concept presented in [18,19,20,21,22,23], key RF systems and subsystems of RF devices and 5G systems can be replaced by corresponding RF MEMS devices to increase their performance, as well as reduce hardware complexity, for example, by combining several functions into one device.

Currently, among all classes of RF switching devices, active semiconductor switches based on PIN-diodes (the diode with a wide no doping intrinsic semiconductor region (i) between a p-type semiconductor and an n-type semiconductor region) currently are the most popular. At frequencies up to 10 GHz, they have almost no equal switching speed, simplicity of circuit solutions and cheapness. However, starting from 8 GHz, PIN-diode switches are not able to provide high isolation in the closed position of the switch, even when cascading. This is due to the increasing influence of the barrier capacitance, which for commercially available PIN diodes is about 0.2–0.8 pF. In addition, PIN diodes cause noticeably greater insertion loss in the open state and at lower frequencies. This problem is partially solved by the use of active semiconductor RF switches based on field-effect transistors (FETs) but, at frequencies up to 40 GHz, coaxial switches are mainly used, which have good RF parameters but are extremely expensive. At frequencies up to 1 GHz, high-frequency relays are sometimes used, but their disadvantages include a low switching speed of at least 5 ms and high power consumption, as well as high weight and size characteristics. RF MEMS switches can effectively replace PIN-diode analogs at frequencies up to 10 GHz and compete with transistor and coaxial switches in the range up to 40 GHz or more [24].

RF MEMS switches are classified according to the actuation mechanism, the movement of the suspended parts, the type of contact, the type of mechanism used and the electrical configuration. Today electrostatic RF MEMS switches are the most common among micromechanical switches. This is due to the almost zero power consumption in the switched on state, the small size of the element, the compatibility of the device manufacturing process with the technological processes of manufacturing ICs using silicon technology and technology based $A^{\mathrm{III}}B^{V}$ elements, the relative ease of manufacture and the short switching time. Electrostatic RF MEMS switches are divided into two types: first—cantilever and second—membrane switches with metal–metal and metal–dielectric–metal contacts, respectively. The second type or capacitive RF MEMS switch is more common when designing RF microswitches [25,26]. Capacitive RF MEMS switches have some advantages over resistive RF MEMS switches, as they are characterized by a lower pull-down voltage value and a short switching time due to the possibility of designing movable suspended structures with a low stiffness coefficient. Furthermore, capacitive RF MEMS switches are characterized by greater reliability and are subject to fewer failure mechanisms. In addition, for the design of RF MEMS switches of medium and low power, switches with a capacitive contact type have an important advantage—the ability to design switch designs with a low value of the pull-down voltage [24,27,28,29,30,31,32].

In capacitive RF MEMS switches, the movable electrode of the design is a metal membrane suspended from the anchor areas by elastic suspensions. The dielectric layer is applied to the signal line of the coplanar waveguide (CPW). Thus, a capacitor with a metal–dielectric layer–metal plate is formed. The main problem of this approach is the requirement of a higher capacitance ratio in the down-state and up-state in order to have a good RF response. In addition, 5G mobile network devices will require higher compactness or a small form factor of discrete RF devices. A number of capacitive RF MEMS switches with a good RF response are reported in the literature, but these RF microswitches are characterized by a high actuation voltage (in a number of scientific works ∼20…50 V) and occupied area on the chip (more 1…2 ${\mathrm{mm}}^{2}$). Another problem of capacitive RF MEMS switches is the imperfect roughness of the contacting layers, which leads to a decrease in the capacitance ratio of this type of RF MEMS switches [33,34]. A decrease in the capacitance ratio, in turn, leads to a shift in the resonant frequency from the required one and a decrease in the isolation value in the closed state of the RF MEMS switch. There are a number of studies that have been conducted to achieve a high value of the capacitance ratio of capacitive RF MEMS switches and a low value of the pull-down voltage. In [35], a design of a capacitive RF MEMS switch using a ceramic dielectric layer with a high permittivity is proposed. In [36], a high capacitance ratio was achieved by using the curved design of the metal membrane of the capacitive RF MEMS switch. Another method used to achieve a high value of the capacitance ratio is to increase the air gap between the metal membrane and the dielectric layer [37,38]. However, there are some obvious disadvantages of these methods, which are that the charge problem of the dielectric layer becomes more significant the smaller the thickness of the applied dielectric layer and the electromechanical parameters of the switch change when the air gap changes. In this regard, the methods used in [35,36,37,38] are not suitable.

Meanwhile, several works have proposed some approaches for obtaining a high value of the capacitance ratio, which consists of the use of dielectric materials with a high permittivity. Such dielectric materials include: Hf$O_{2}$ ($\varepsilon_{r}$ = 20) [39], STO ($\varepsilon_{r}$ = 30–120) [35,40], ${\mathrm{Ta}}_{2}O_{5}$ ($\varepsilon_{r}$ = 32) [41], (Ba, Sr) Ti$O_{3}$ ($\varepsilon_{r}$ > 200) [42], PZT ($\varepsilon_{r}$ = 190) [43] and metal oxide dielectrics with high dielectric characteristics [43]. As a result, the value of the capacitance ratio of capacitive RF MEMS switches is more than 100 [43,44]. However, the value of the capacitance ratio is limited by the minimum thickness of the dielectric layer, the maximum value of the dielectric constant and the maximum value of the air gap between the movable switch electrode and the RF signal line CPW. To eliminate the disadvantages described above, it is necessary to combine constructive methods for designing capacitive RF MEMS switches with methods of using high-k dielectric materials (with a high permittivity) to obtain RF microswitches with a high capacitance ratio and a good RF response.

The purpose of this article is to design, manufacture and study the designs of high-performance capacitive RF MEMS switches with a high capacitance ratio, eliminating the disadvantages listed above, using the developed methodology for designing high-performance capacitive RF MEMS switches for certain applications and devices based on the operating resonant frequency or frequency range, eliminating the shortcomings of traditional methodologies and approaches for designing microelectromechanical devices. Moreover, this approach to increasing the value of the capacitance ratio consists of using the design of a floating metal movable electrode without restrictions of the minimum thickness of the dielectric layer and the minimum value of the air gap, as well as using the high-k dielectric material. A floating metal layer is included in the design of the RF switch to exclude a decrease in the capacitance ratio due to residual stresses and imperfections in the surface roughness after technological operations of manufacturing the device. In addition, the capacitive RF MEMS switch developed for use as a discrete passive component in RF devices of 5G mobile networks should be characterized by a good electromechanical, dynamic and RF response, low insertion contact resistance in the closed state, high reliability and a small form factor.

## 2. Methodology of Designing High-Performance Capacitive RF MEMS Switches

The RF performance characteristics of RF MEMS switches are an important factor along with the electromechanical characteristics not only in the scenario of their use in RFFE devices of 5G mobile networks. The traditional optimization of the MEMS device design available in the latest literature is mainly based on independent optimizations of the device design with one physics area corresponding to each physical process or phenomenon, and then a logical combination of independently optimized designs provides the overall optimal design of the MEMS device, as shown in Figure 1.

Traditional optimization methods are limited to the response of one output parameter of the MEMS device and are based on optimization in one area of physics and a logical combination of individual designs, which provides overall optimization [45,46,47,48,49,50,51,52,53]. However, these methods are not an effective approach for MEMS devices, in particular RF MEMS devices, which are complex structures with multiphysical interaction. In this article, the methodology of optimization of RF MEMS switches based on multiphysical modeling of finite element methods (FEM) was considered, as shown in Figure 2. The basis of this approach is the development of an RF MEMS switch for transmitting an RF signal based on a resonant frequency for certain applications and devices. The RF MEMS switch consists of five levels. Optimization of each level, starting from the CPW and substrate to the additional fixed down capacitor (upper level) is also referred to in the literature as the “bottom-up” approach [52]. In this paper, the methodology under consideration is a suitable tool for the development of the capacitive RF MEMS switches using surface micromachines technology.

At the same time, various levels of the design process provide for analytical calculations and modeling using the FEM with CAD software tools, checking for compliance with key design parameters. If the key design parameters are not met at one of the design stages, the level returns to the previous stage to achieve the required values. Ultimately, an optimized design of a capacitive RF MEMS switch with high performance is achieved, designed to operate in a certain frequency range.

The proposed optimization method is used to design RF MEMS switches with a capacitive contact type for a defined RF device and application area based on the design for a defined resonant frequency, as shown in Figure 2. The proposed methodology is based on a “bottom-up” approach, which uses analytical calculations and multiphysical modeling using the FEM to optimize the parameters of the RF MEMS switch design. At the same time, the proposed methodology can be adjusted accordingly for the design process of RF MEMS switches with a resistive contact type.

Initially, when designing RF MEMS switches with a capacitive contact type, the type of RF waveguide is selected and, for designing RF MEMS switches with a capacitive contact type, the priority type of RF waveguide is the CPW for many reasons: firstly, when using the CPW, it is possible to use thick dielectric substrates; secondly, since the grounding lines are located on the front side of the substrate, they can be used as control electrodes; thirdly, when using the CPW, it is technologically more accessible and makes it possible to manufacture RF MEMS switch designs of various configurations and complexity.

At the first stage, the most suitable material of the conductive lines of the CPW and the substrate is selected. When choosing the material of the main conductive layer of the projected CPW, as a rule, they focus on a low electrical resistivity ρ, a high value of the thermal conductivity K and a coefficient of thermal expansion $\alpha_{T}$. Table 1 presents the properties of the most common materials used for the design of the CPW.

When choosing a substrate material, it is necessary to follow a certain methodology that allows you to characterize the appropriate material for the desired performance of the RF MEMS switch, depending on its properties (mechanical, EM and thermal). Different device designs require a specific set of these characteristics. The methodology of material selection includes five stages, as shown in Figure 3.

Design requirements for a structural component are derived based on functions, goals and constraints. The next step of the methodology shows that a wide choice of materials is narrowed, firstly, by applying property constraints that allow a narrowing of the spectrum of materials that cannot meet the requirements of the device design. Further narrowing is achieved by applying material indexes and ranking materials based on their ability to provide the best performance for a given device design. The material index is a set of material properties that maximize the performance of the device design for a given requirement. These material indices are derived from the design requirements for the design of the device by analyzing functions, goals and limitations. A material performance index is a group of material properties that directly affects some aspects of the performance of the device design. Material selection using performance indicators is best achieved by plotting one material property on each axis of the material selection diagram. The design of the device is usually determined by three main parameters: functional requirements, geometric properties and material properties. The performance of the structural elements of the device is described as: P = $f_{1}\left(F\right)\cdot f_{2}\left(G\right)\cdot f_{3}\left(M\right)$. The performance of the structural elements of the device P is described by the individual functions F, G and M. Thus, the optimal subset of materials can be determined by a single functional requirement. For all F and G, performance can be optimized by optimizing the corresponding material metrics. This optimization is traditionally performed using graphs with axes corresponding to different material indices or material properties.

To use RF MEMS switches in the microwave region of RF wavelengths, substrates made of highly resistive material with high temperature stability are used. The resistivity of the material should be ρ≥5000 Ω·m. Based on these conditions, the most suitable substrate materials are: quartz, sapphire, glass, ferrite, granite, GaAs, ${\mathrm{Al}}_{2}O_{3}$, AlN, BeO, GaN, InP, LTCC, SiC and highly resistive silicon substrate (HR Si). At UHF (from 1 GHz and above), the size of the chip becomes comparable to the order of the length of the radio-wave, which affects its performance. For this reason, the substrate material must have certain characteristics:-High dielectric constant, $\varepsilon_{d}$;-Low dielectric loss tangent, $\tan\delta_{e}$;-High resistivity, ρ;-High thermal conductivity, K.

It should be noted that it is important to avoid the emission of an EM field into the air, therefore it is necessary to use substrates with high dielectric permittivity with recommended values exceeding 10 kΩ-cm for the EM field to be mainly concentrated inside the dielectric of the substrate. For this reason, the height of the dielectric substrate and the relative permittivity $\varepsilon_{d}$ of the material are the main parameters for designing the CPW signal line, as well as for studying the distribution of electric and magnetic fields. In addition to electromagnetic characteristics, mechanical and physical properties are important: low surface roughness, purity of the material and constant thickness.

Table 2 present the main parameters for choosing the material of the substrate.

At the beginning of the analysis of the choice of substrate material, ferrite and granite are excluded because they are characterized by low thermal conductivity and dielectric strength. The choice of material is determined by three main parameters:-Functional requirements;-Geometric properties;-Properties of the material.

The main material indexes M when choosing the substrate material for RF MEMS switches are:-The first material index $M_{1}$ = $\varepsilon_{\mathrm{eff}}$ is associated with dielectric loss in the CPW or effective permittivity $\varepsilon_{\mathrm{eff}}$;-The second material index $M_{2}$ = $\tan\delta_{e}$ is related to the tangent of dielectric loss;-The third material index $M_{3}$ = $\alpha_{\Delta }$ and the first performance index $P_{1}$ = $f\left(\varepsilon_{\mathrm{eff}},\tan\delta_{e},\alpha_{\Delta }\right)$ is the value of dielectric loss or attenuation;-The fourth material index $M_{4}$ = ρ is the loss of RF power in the substrate, while, since the loss level is directly proportional to the electrical resistivity of the substrate material, the second performance index $P_{2}$ = $f\left(\rho \right)$ is electrical loss;-The result of the electrical and thermal resistances of the substrate material induces heating of the substrate material, which means that the fifth material index $M_{5}$ = $\frac{1}{K}\rho$ and the third performance index $P_{3}$ = $f\left(\frac{1}{K}\rho \right)$ are thermal residual stresses.

Thus, based on their analysis of the choice of suitable substrate material for capacitive RF MEMS switches, the most acceptable materials, based on the material and performance indices, are sapphire and high-purity ${\mathrm{Al}}_{2}O_{3}$, which have the highest thermal conductivity K, the lowest value of the tangent of the dielectric loss angle $\tan\delta_{e}$ and the highest value of the relative permittivity $\varepsilon_{d}$. At the same time, the roughness of the sapphire surface is better.

Then the dimensions of the CPW are optimized using the skin depth effect $\delta_{\mathrm{CPW}}$ of the screen layer and the criteria for choosing the substrate. At the same time, surface waves passing through the CPW are studied, and the thickness of the substrate is selected by the resonant and upper cut frequency. The CPW and the substrate are optimized so as to have a characteristic resistance *Z*${}_{0}$ of 50 Ω. The calculation of the wave resistance $Z_{\mathrm{CPW}}$ of the CPW is carried out using the method of conformal transformations and partial capacitances. It is assumed that a quasi-static transverse EM (TEM) wave propagates in such a structure, which differs from the TEM wave in that the transverse components of the EM field in it are significantly larger than the longitudinal ones (there are no longitudinal components in the TEM wave). In addition, in this case, the CPW has grounding planes of finite width, a substrate of finite thickness and conductors of finite thickness. Then the concentrated parameters of the CPW are extracted in order to understand the level of attenuation, insertion loss $S_{11}^{\mathrm{CPW}}$ and reflection loss $S_{12}^{\mathrm{CPW}}$ during RF signal transmission. The value of the total attenuation α in the CPW is determined by two components: (1) attenuation due to loss in the dielectric substrate $\alpha_{d}$ and (2) attenuation in conductors $\alpha_{c}$ (RF signal line and grounding lines). The level of $S_{11}^{\mathrm{CPW}}$ and $S_{12}^{\mathrm{CPW}}$, the value of α in the CPW, is determined by two components: (1) $\alpha_{d}$ and (2) $\alpha_{c}$. In this case, the attenuation corresponding to dielectric loss depends more on the constant dielectric parameters of the substrate used: the relative permittivity of the substrate $\varepsilon_{r}$; the tangent of the dielectric loss angle $\tan\delta_{e}$; and the effective dielectric permittivity $\varepsilon_{\mathrm{eff}}$, while, for most dielectric substrate materials, the $\tan\delta_{e}$ remains constant with increasing frequency of the RF signal. However, the $\alpha_{d}$ increases linearly with increasing frequency and is determined by the sequential resistance per unit length of the RF signal line $R_{c}$ and the distributed sequential resistance per unit length of the ground lines $R_{g}$ of the CPW. Furthermore, in the designed topology of the CPW, inductive tuning is performed using special areas in the grounding lines symmetrically on both sides relative to the RF signal line. These inductive regions in the CPW are designed to place fixed electrostatic activation electrodes symmetrically on both sides relative to the RF signal line, and also by further EM optimization allow the operation of the RF MEMS switch at the required resonant frequency to be achieved and increase the amount of isolation.

At the next stage, the design of the movable membrane of the capacitive RF MEMS switch, the profile and the number of through-perforated holes in it are designed to reduce the isothermal damping coefficient, increase the switching speed and reduce the time of the closing and opening operation. In this case, the first step of this stage is the selection of a suitable material for the movable membrane of the switch, as well as, subsequently, elastic suspension elements on which the movable membrane is fixed to the anchor areas. Table 3 presents the main parameters for choosing the material of the movable membrane and elastic suspension elements.

The selection of the most suitable material is made according to the methodology presented earlier:-The first material index $M_{1}$ = $\sqrt{E}$ is related to the value of the Young’s modulus of the material;-The second material index $M_{2}$ = $\left(\nu \right)$ is related to the value of the Poisson’s ratio of the material;-The third material index $M_{3}$ = $\left(\alpha_{T}\right)$ is related to the value of the coefficient of thermal expansion of the material;-The first performance index $P_{1}$ = $f\left(E,\nu ,\alpha_{T}\right)$ is related to the value of the control voltage;-The fourth material index $M_{4}$ = $\left(\rho \right)$ and the second performance index $P_{2}$ = $f\left(\rho \right)$ are related to the value of the electrical resistance of the material and the value of the RF loss that occurs;-The fifth material index $M_{5}$ = $\left(\frac{1}{K}\rho \right)$ and the third performance index $P_{3}$ = $f\left(\frac{1}{K}\rho \right)$ are related to the thermal conductivity of the material and thermal residual stresses.

According to the data obtained, a material with high values of Poisson’s ratio ν and the coefficient of thermal expansion $\alpha_{T}$ is most suitable for minimizing the value of the control voltage. There is a compromise between gold and aluminum. At the same time, gold demonstrates a higher value of Poisson’s ratio ν at a lower value of the coefficient of thermal expansion $\alpha_{T}$ and aluminum demonstrates a higher value of the coefficient of thermal expansion $\alpha_{T}$ at a lower value of Poisson’s ratio ν. A material with a low value of Young’s modulus E and a high value of Poisson’s ratio ν helps to reduce the value of the control voltage, while gold and aluminum are the most suitable materials. Aluminum and gold demonstrate a minimum value of Young’s module E and electrical resistivity ρ to ensure a minimum value of RF loss. In addition, gold, copper and aluminum provide a minimum amount of thermal residual stress at a low value of electrical resistivity ρ, with a high value of thermal conductivity K of the material. Thus, based on their analysis of the choice of a suitable material for a movable membrane and elastic suspension elements for the design of capacitive RF MEMS switches, aluminum is the most acceptable material, based on material indexes M and performance indexes P, as well as economic feasibility.

Next, the constitution of the movable membrane of the capacitive RF MEMS switch is designed directly; the geometric dimensions are calculated and optimized based on the calculation of key parameters such as membrane resistance $R_{b}$ and inductance $L_{b}$. The thickness is determined based on the calculation of the skin depth effect $\delta_{b}$. The calculated resistance of the membrane $R_{b}$ should not exceed 0.01–0.1 Ω, since a lower resistance provides lower values for insertion loss $S_{11}$ and reflection loss $S_{12}$ in a given frequency range with the RF MEMS switch open state and in the closed state the RF signal will flow through the membrane through the RF signal line of the CPW; therefore, in this case, the equivalent resistance will consist of two components: $R_{b}+R_{\mathrm{CPW}}$. At the same time, in the closed state, the lower resistance of the membrane $R_{b}$ provides a greater amount of isolation and a smaller amount of return loss.

To reduce the damping coefficient and increase the switching speed of the RF MEMS switch in the movable membrane, perforated holes are provided in the design. The area of these holes can be up to 60% of the total area of the membrane. The holes in the membrane also allow part of the residual stresses in the movable structure after the technological process of deposition of the layer to be reduced, lead to a decrease in the value of Young’s modulus E by approximately 25–30% and also allow the mass of the movable structure to be reduced, which in turn leads to a higher mechanical resonance frequency and greater switching speed. Then the coefficient of isothermal damping β occurring at a constant temperature is determined. The process of compression of the movable membrane during the electrostatic activation of the RF MEMS switch occurs in different ways and mainly depends on the chosen design, i.e., on the directions and the possibility of lateral removal of the compressed air. The specified boundary condition, in this case, is the case when all sides under the movable membrane are open, which is the most common option when designing capacitive RF MEMS switches. The damping parameters in the considered oscillatory system have a direct effect on the Q-factor or quality factor of the system. The Q-factor is one of the most important parameters of the RF MEMS switches and can be explained as the ratio of stored energy to the energy dissipated per cycle at a resonant frequency. A higher Q-factor indicates a lower rate of energy loss compared to the accumulated energy of the resonant circuit. The Q-factor is also influenced by thermoelastic damping ξ, which occurs in the RF MEMS switch when switching RF signals due to irreversible heat flow caused by local temperature gradients acting on the movable membrane. This stage is accompanied by an analytical calculation of electrical, mechanical, thermomechanical and EM parameters, as well as appropriate modeling using the FEM using CAD software tools.

Then, the design of fixed down actuation electrodes designed for the electrostatic activation of the RF MEMS switch is carried out. The material for the design is a conductive material defined for the design of the CPW. For electrostatic activation of the projected capacitive RF MEMS switch, two fixed down electrodes are required, which are placed symmetrically under the movable membrane on both sides of the RF signal line of the CPW with some air gap to prevent the RF signal line of the CPW from closing with each of the fixed down electrodes. At the same time, a thin passivating layer of $\mathrm{Si}O_{2}$ or ${\mathrm{Si}}_{3}N_{4}$ is deposited on the surface of the fixed down electrodes. The design of contact pads and connecting conductive lines is carried out to supply the control voltage to the fixed down electrodes, followed by passivation with a layer of $\mathrm{Si}O_{2}$ or ${\mathrm{Si}}_{3}N_{4}$ for electrical isolation of the control voltage from the RF signal and to prevent a short circuit.

At the next stage, the elastic suspension elements of the movable membrane are designed. At the same time, the design of the elastic suspension must have a low stiffness coefficient in the required direction to reduce the value of the control voltage and at the same time be characterized by a sufficiently high stiffness coefficient in other directions, since this reduces the sensitivity of other directions to the action of cross-acceleration. Furthermore, different types of elastic suspensions are characterized by different inductance, which is introduced in addition to the movable membrane.

The next stage of designing the capacitive RF MEMS switch is the development of an additional fixed down capacitor with metal–dielectric–metal (MIM) plates. The dielectric is a dielectric material with a high permittivity $\varepsilon_{r}$—high-k-dielectric. The design of this structural element makes it possible to increase the capacitance ratio of capacitive RF MEMS switches and their EM parameters, eliminating the disadvantages of the classical approach of designing capacitive RF MEMS switches, which were mentioned earlier in the problem statement. The first step is to choose the material for forming the MIM capacitor. The main parameters in this case are:-Dielectric constant, $\varepsilon_{r}$;-Electrical resistivity, ρ;-Thermal conductivity, K;-Coefficient of thermal expansion, $\alpha_{T}$;-Young’s modulus, E.

The selection of the material takes place using the methodology presented earlier:-The first material index is related to the value of the dielectric constant $\varepsilon_{r}$ of the material $M_{1}$ = $\left(\frac{1}{\sqrt{\varepsilon_{r}}}\right)$, while the first performance index $P_{1}$ = $f\left(\frac{1}{\sqrt{\varepsilon_{r}}}\right)$ is related to the value of the control voltage;-The second material index $M_{2}$ = $\left(\rho \cdot \varepsilon_{r}\right)$ is associated with the value of the electrical resistance and the value of the dielectric constant, while the second performance index $P_{2}$ = $f\left(\rho ,\varepsilon_{r}\right)$ is associated with the electric charge of the dielectric layer τ;-The third material index $M_{3}$ = $\left(E\right)$ is associated with the value of Young’s modulus;-The fourth material index $M_{4}$ = $\left(\alpha_{T}\right)$ is associated with the value of the coefficient of thermal expansion;-The fifth material index $M_{5}$ = $\left(K\right)$ is associated with the value of the thermal conductivity;-The third performance index $P_{3}$ = $f\left(E,\alpha_{T},K\right)$ is related to the efficiency of thermal stress relaxation of the RF MEMS switch Δσ;-The sixth material index $M_{6}$ = $\left(\varepsilon_{r}\right)$ is related to the value of the dielectric constant, while the fourth performance index $P_{4}$ = $f\left(\varepsilon_{r}\right)$ is related to the value of the capacitance ratio $C_{r}$ and, accordingly, the EM parameters obtained.

Table 4 shows the main parameters for choosing the material of the dielectric layer of the MIM capacitor.

According to the data obtained, the dielectric material must have a very high electrical resistivity ρ and a high permittivity $\varepsilon_{r}$. Thus, from the given list of materials, it can be concluded that the $\mathrm{Ti}O_{2}$ dielectric material has the highest value of dielectric permittivity and the value of resistivity exceeds the threshold values. However, sometimes high-k dielectrics cause problems with adhesion in capacitive RF MEMS switches due to the presence of a dielectric charge but this problem is solved constructively in this design methodology: there is no direct contact of the movable membrane with a high-k dielectric; a high-k dielectric allows you to get an additional MIM capacitor to increase the resulting capacitance ratio with which the movable membrane contacts during electrostatic activation of the switch.

At the same time, the value of thermal conductivity K is higher in dielectric materials AlN, ${\mathrm{Ta}}_{2}O_{5}$ and ${\mathrm{Si}}_{3}N_{4}$, which is desirable for RF applications with high power, where any of the structural elements of the RF MEMS switch design may undergo irreversible damage due to overheating. For applications with medium and low RF signal power $\mathrm{Ti}O_{2}$ can be selected.

Then, the design and calculation of the capacity of an additional fixed MIM capacitor located on the RF signal line of the CPW is carried out directly, which is connected in series with the main variable capacitor, which is formed by the upper metal lining of the MIM capacitor, an air gap and a movable metal membrane—a metal–air–metal (MAM) capacitor. At the same time, this approach of increasing the capacitance ratio of capacitive RF MEMS switches reduces the area of the movable membrane and the area of the main dielectric layer and, accordingly, reduces the switching speed and the form factor of the switch. The calculation of the resulting capacitance value in the up-position and the down-position of the movable electrode, and the calculation of key EM parameters are performed. In case of non-compliance with the specified design values, optimization and recalculation of the fixed additional MIM capacitor is performed. This stage is accompanied by an analytical calculation of EM parameters, as well as appropriate EM and transient thermal modeling using the FEM using CAD software tools.

## 3. Proof of Methodology

### 3.1. Design of Structures

In accordance with the proposed optimization methodology in Figure 2, two designs of capacitive RF MEMS switches were developed [54,55,56,57,58]. The first design is a classic arrangement of elements for the design of capacitive RF MEMS switches—RF MEMS switch (A), in which the movable membrane is located perpendicular to the RF signal line of the CPW with a certain amount of air gap $g_{0}$. In the second design, the movable membrane is part of the RF signal line of the CPW—the inline capacitive RF MEMS switch—RF MEMS switch (B). At the same time, in both designs of capacitive RF MEMS switches, the movable membrane, during electrostatic activation, contacts the upper metal layer of an additional fixed down MIM capacitor. The additional fixed down MIM capacitor is electrically connected in series with a shunt variable MAM capacitor, which is formed by the upper metal layer of the MIM capacitor, the movable membrane and an air gap $g_{0}$ between them. The MIM capacitor is connected in series with the MAM capacitor when the movable membrane is in the up-state. In the case when the movable metal membrane is in the down-state, the MAM capacitor is replaced by an equivalent resistance in the electrical circuit—a hybrid type of contact, as shown in Figure 4.

Figure 4 schematically shows a one-dimensional model of the motion of a movable membrane of the developed designs of the RF MEMS switch. The control voltage is applied between the movable membrane (positive potential) of the RF MEMS switch structure connected to the ground lines of the CPW by means of anchor areas and the fixed down actuation electrodes (negative potential). In this case, the movable membrane of the RF MEMS switch is affected by the force of electrostatic interaction, which is balanced by the mechanical force of elasticity, depending on the coefficient of elasticity of the elastic elements of the suspension. The balance of forces exists as long as the mechanical elastic force, which is a linear function, can compensate for the growth of the electrostatic interaction force, which varies according to the quadratic law. At some point, the magnitude of the mechanical elastic force cannot compensate for the increase in the force of the electrostatic interaction and the movable membrane of the RF MEMS switch falls on the fixed down actuation electrodes, transferring the RF MEMS switch to a closed state. In the presented designs of RF MEMS it switches to an additional fixed down MIM capacitor. When the electric potential applied between the movable membrane and the fixed down actuation electrodes is removed, the movable membrane returns to its original position due to the action of the mechanical elastic force caused by the elastic suspension elements, transferring the RF MEMS switch to the open state.

The characteristic resistance of the CPW $Z_{0}$.$Z_{\mathrm{bl}}\left(\alpha ,\beta l\right)$ denotes the characteristic resistance of the RF signal line between the wave ports and the edge of the movable membrane; α is the transmission attenuation constant and β is the electrical length of the RF signal line.

The isometric 3D topology of the developed capacitive RF MEMS switches is shown in Figure 5. In both presented designs of capacitive RF MEMS switches the upper metal layer of the additional fixed down MIM capacitor and the relatively narrow RF signal line of the CPW are located under the movable membrane of the switch in order to obtain a low capacitance value in the up-state. Changing the size of the dielectric layer ensures that the wave resistance 50 Ω is matched between the RF input of the CPW RF signal line and the connected load.

The design of the first proposed RF MEMS switch (A) consists of the following set of structural elements: the dielectric substrate; the metal conductive layer forming the CPW (RF signal line and grounding lines located symmetrically on both sides); fixed down electrodes for electrostatic activation and their control voltage lines with contact pads for supplying a positive potential of the control voltage, as well as passivation layers on them; the high-k dielectric layer located locally on the surface of the RF signal line of the CPW, the metal conductive layer located on the surface of the high-k dielectric layer and repeating its topological pattern with some indentation, forming together an additional fixed MIM capacitor $\left(C_{{\mathrm{MIM}}_{1}}\right)$; and the movable metal membrane suspended over an additional fixed MIM capacitor by means of four metal conductive elastic suspension elements that are fixed to four metal conductive anchor areas locally located on the grounding lines of the CPW.

The difference between RF MEMS switch (B) and the first is the location and electrical connection of an additional fixed MIM capacitor, as well as the location and electrical connection of a movable metal membrane. In this case, the metal conductive layers of the grounding lines of the CPW have a direct electrical connection by means of a metal conductive layer between them. The high-k dielectric layer is locally located on the surface of this metal conducting layer, on its surface a metal conducting layer repeating its topological pattern also with some indentation, forming together an additional fixed MIM capacitor $\left(C_{{\mathrm{MIM}}_{2}}\right)$. The movable metal membrane is suspended above an additional fixed MIM capacitor by means of two metal conductive elastic suspension elements, which are fixed to two metal conductive anchor areas locally located on the RF signal line of the CPW in such a way that the movable metal membrane is part of the RF signal line of the CPW.

#### 3.1.1. Designs of the Coplanar Waveguide

The basic structure of the CPW comprises a symmetric arrangement with signal strip width W and equal longitudinal gap G. Finite-ground CPW (FG-CPW) is used in this design where the grounding lines are not shared by two or more lines and hence result in a lower coupling of the adjacent lines. CPW are preferred as they allow easy surface mounting of the devices (here RF MEMS switches), and the reduced dispersion and radiation loss. Figure 6 schematically shows the two different dimensions of the CPW, namely, CPW (A) and CPW (B) as used in this design of the capacitive RF MEMS switches. Figure 7 schematically shows the main geometric dimensions of CPW (A) and CPW (B).

In this paper, CPW (A) and CPW (B) were chosen for the capacitive RF MEMS switch design, since in this case it is possible to use thick dielectric substrates (for example, ${\mathrm{Al}}_{2}O_{3}$—500 μm). Since the grounding lines are located on the front side of the substrate, they can be used as control electrodes. Thus, the use of CPW in the design of RF MEMS switch designs is technologically more accessible and makes it possible to manufacture structures of various configurations and complexity. The CPW for transmitting a UHF RF signal is formed from a symmetrically arranged RF signal line with a width W relative to the ground lines and an equal longitudinal gap G between them.

The dimensions for CPW (A) and CPW (B) are specified in Table 5.

The calculation of the characteristic resistance $Z_{0}$ of the CPW is carried out using the method of conformal transformations and partial capacitances, while it is assumed that a quasi-static TEM wave propagates in such a structure and also, in this case, a CPW with grounding lines of finite width, a dielectric substrate of finite thickness and conductors of finite thickness is considered. Using this approximation, the phase velocity $V_{\mathrm{ph}}$, effective permittivity $\varepsilon_{\mathrm{eff}}$ and characteristic resistance $Z_{0}$ are determined by Equation (Equation 1):(1)$$\varepsilon_{\mathrm{eff}}=\frac{C}{C_{0}};\;V_{\mathrm{ph}}=\frac{c^{\prime }}{\sqrt{\varepsilon_{\mathrm{eff}}}};\;Z_{0}=\frac{1}{c^{\prime }V_{\mathrm{ph}}}$$
where $c^{\prime }$ is the speed of light in free space; C is the capacitance per unit length of the line and $C_{0}$ is the capacitance per unit length of the line in the absence of a dielectric substrate.

To determine the constituent capacitances, it is assumed that the boundaries of the dielectric layers run along the electric field lines. In this case, the magnetic walls can be placed along the electrical boundaries without disturbing the fields and the capacity of the CPW can be divided into partial capacities. Under this assumption the capacitance *C* of the CPW can be written as a superposition of two partial capacities using Equation (Equation 2):(2)$$C=C_{0}+C_{1}$$

The capacitance $C_{0}$ in the absence of a dielectric substrate is determined by Equation (Equation 3):(3)$$C_{0}=4\varepsilon_{0}\frac{K\left(k^{\prime }\right)}{K\left(k\right)}$$

To determine the capacitance $C_{1}$, it is assumed that the electric field exists only in a dielectric layer with a thickness $h_{1}$ and relative permittivity $\varepsilon_{r_{1}}-1$. In this case, the capacitance $C_{1}$ is determined by Equation (Equation 4):(4)$$C_{1}=2\varepsilon_{0}\left(\varepsilon_{r_{1}}-1\right)\frac{K\left(k_{1}^{\prime }\right)}{K\left(k_{1}\right)}$$
where $k_{1}$ and $k_{1}^{\prime }$ are determined by Equation (Equation 5):(5)$$\begin{array}{c} k_{1}=\frac{\sin h\left(\pi c/{2\mathrm{ch}}_{1}\right)}{\sin h\left(\pi b/{2\mathrm{ch}}_{1}\right)}\sqrt{\frac{{\sin h}^{2}\left(\pi {b/2h}_{1}\right)-{\sin h}^{2}\left(\pi {a/2h}_{1}\right)}{{\sin h}^{2}\left(\pi {c/2h}_{1}\right)-{\sin h}^{2}\left(\pi {a/2h}_{1}\right)}}\\ k_{1}^{\prime }=\sqrt{1-k_{1}^{2}}=\frac{\sin h\left(\pi a/{2\mathrm{ch}}_{1}\right)}{\sin h\left(\pi b/{2\mathrm{ch}}_{1}\right)}\sqrt{\frac{{\sin h}^{2}\left(\pi {c/2h}_{1}\right)-{\sin h}^{2}\left(\pi {b/2h}_{1}\right)}{{\sin h}^{2}\left(\pi {c/2h}_{1}\right)-{\sin h}^{2}\left(\pi {a/2h}_{1}\right)}} \end{array}$$

The thickness of the substrate is 500 μm ≫ the thickness of the CPW line (6 μm). Hence, for an infinitely thick substrate $k_{s}=k$. The equation for the effective permittivity $\varepsilon_{\mathrm{eff}}$ is reduced to Equation (Equation 6):(6)$$\varepsilon_{\mathrm{eff}}=\frac{C}{C_{0}}\frac{1}{2}\left(\varepsilon_{r}+1\right)\frac{K\left(k\right)}{K\left(k^{\prime }\right)}\frac{K\left(k_{1}^{\prime }\right)}{K\left(k_{1}\right)}$$

Thus, the characteristic resistance $Z_{0}$ is given by Equation (Equation 7):(7)$$Z_{0}=\frac{30\pi }{\sqrt{\varepsilon_{\mathrm{eff}}}}\cdot \frac{K\left(k\right)}{K\left(k^{\prime }\right)}$$

Figure 8 and Figure 9 shows the distribution of electric E and EM H fields in the developed design of CPW (A) and CPW (B). Figure 10 shows a graph of the distribution of the characteristic resistance $Z_{0}$ and the value standing wave ratio (VSWR) of CPW (A) and CPW (B) from the frequency of the RF signal.

In the conformal mapping method the total attenuation in the CPW is determined by a superposition of two factors: attenuation due to loss in the dielectric substrate and attenuation in conductors (RF signal line and grounding lines). The attenuation corresponding to the dielectric loss is determined by Equation (Equation 8):(8)$$\alpha_{d}=\frac{\pi }{\lambda_{0}}\frac{\varepsilon_{r}}{\sqrt{\varepsilon_{\mathrm{eff}}}}q\tan\delta_{e}$$
where $\lambda_{0}$ is the wavelength in free space and q is the filling factor determined by Equation (Equation 9):(9)$$q=\frac{1}{2}\frac{K\left(k^{\prime }\right)}{K\left(k_{1}^{\prime }\right)}\frac{K\left(k_{0}^{\prime }\right)}{K\left(k_{0}\right)}$$

The terms $K\left(k_{1}\right)$ and $K\left(k_{0}\right)$ are complete elliptic integrals of the first kind with modules $k_{1}$ and $k_{0}$, which are defined by Equations (Equation 10) and (Equation 11):(10)$$\begin{array}{c} k_{1}=\frac{\sin h\left(\pi S/4h_{1}\right)}{\sin h\left[\pi \left(S+2W\right)/4h_{1}\right]}\\ k_{0}=\frac{S}{S+2W} \end{array}$$
(11)$$k_{1}^{\prime }=\sqrt{1-k_{1}^{2}};\;k_{0}^{\prime }=\sqrt{1-k_{0}^{2}}$$

For various dielectric materials, the tangent of the dielectric loss angle $\tan\delta_{e}$ remains constant with increasing frequency, but the attenuation corresponding to dielectric loss $\alpha_{d}$ increases linearly with increasing frequency. Attenuation in conductors is determined using Equation (Equation 12):(12)$$\alpha_{c}=\frac{R_{c}+R_{g}}{2Z_{0}}$$
where $R_{c}$ is the serial resistance per unit length of the RF signal line, defined by Equation (Equation 13), and $R_{g}$ is the distributed serial resistance per unit length of the grounding lines, defined by Equation (Equation 14):(13)$$R_{c}=\frac{R_{s}}{4S\left(1-k_{0}^{2}\right)K^{2}\left(k_{0}\right)}\left[\pi +\ln\left(\frac{4\pi S}{t}\right)-k_{0}\ln\left(\frac{1+k_{0}}{1-k_{0}}\right)\right]$$
(14)$$R_{g}=\frac{k_{0}R_{s}}{4S\left(1-k_{0}^{2}\right)K^{2}\left(k_{0}\right)}\left[\pi +\ln\left(\frac{4\pi \left(S+2W\right)}{t}\right)-\frac{1}{k_{0}}\ln\left(\frac{1+k_{0}}{1-k_{0}}\right)\right]$$
where $R_{s}$ is the resistance of the skin layer defined by Equation (Equation 15):(15)$$R_{s}=\frac{1}{\delta \sigma }$$
where σ is the conductivity; δ is the thickness of the skin layer.

In quantitative terms, the thickness of the skin layer is defined as the distance by which the EM field exponentially attenuates to $e^{-1}$ = 0.368 or 36.8% of its value at the air–conductor interface, and determined by Equation (Equation 16):(16)$$\delta =\frac{1}{\sqrt{f\pi \mu \sigma }}$$
where f is the frequency of the RF signal; μ is the permeability of the medium surrounding the conductors; and σ is the conductivity of the metal composing the conductor.

The minimum thickness of the metal conductor for the design of the CPW should be chosen in such a way that it is approximately equal to or exceeds four skin depths $\left(t_{\mathrm{CPW}}\geq 4\delta \right)$ at the lowest operating frequency of the RF device. Thus, the recommended thickness of the developed CPW (A) and CPW (B) is 6 μm for operation at a central resonant frequency of 3.6 GHz and 3.4 GHz.

Thus, the total attenuation in the CPW is determined by Equation (Equation 17):(17)$$\alpha =\left(\alpha_{d}+\alpha_{c}\right)=20\log_{10}e\left(\alpha_{d}+\alpha_{c}\right)$$

The value of the insertion loss into the CPW length $L_{\mathrm{CPW}}$ will be determined by Equation (Equation 18):(18)$$S_{11}=20\log_{10}e\left(\alpha_{d}+\alpha_{c}\right)L_{\mathrm{CPW}}$$

The value of reflection loss in the CPW is determined by Equation (Equation 19):(19)$$S_{12}=10\log_{10}\left|1-{\left(S_{11}\right)}^{2}\right|$$

Figure 11 show the results of modeling the scattering parameters, as well as the results of the associated transient thermal modeling during the passage of a 1W RF signal for the developed CPW (A) and CPW(B).

From the data presented this is also reflected in good scattering parameters: insertion loss $S_{11}$ for CPW (A) is −0.06 dB and for CPW (B) is −0.14 dB; reflection loss $S_{12}$ for CPW (A) is −33.06 dB and for CPW (B) is −24.47 dB.

When an RF signal of the design frequency with a power of 1W is transmitted, the maximum temperature of the CPW (A) structure is $80.44{\;}^{\circ }C$, for CPW (B)—$65.17{\;}^{\circ }C$, which is not the melting or softening temperature of the metal conductive layer and cannot be the cause of its deformation.

#### 3.1.2. Designs of the Movable Membranes

Figure 12 shows the optimized design of the movable membrane of the developed capacitive RF MEMS switches.

The conductivity of the RF signal between the input and output of the RF signal line of the CPW depends on the position of the switching movable membrane: for switch (A)—located above the RF signal line of the CPW; for switch (B)—being part of the RF signal line of the CPW. The various geometric dimensions of the movable membranes of the developed capacitive RF MEMS switches are shown in Table 6.

For both types of developed capacitive RF MEMS switches in the closed state, electric current will flow through the movable membrane through the RF signal line of the CPW. Hence, the equivalent electrical resistance of the developed switches can be given by Equation (Equation 20):(20)$$R_{b}=R_{m}+R_{\mathrm{CPW}}$$
where $R_{m}$ is the resistance due to the movable membrane and $R_{\mathrm{CPW}}$ is the resistance of the CPW central conductor, respectively.

At HF or UHF of the RF signal, skin effects occur due to depth skin $\delta_{b}$. Thus, the area in Equation (Equation 21) is equal to:(21)$$\delta_{m}=\sqrt{\frac{2}{\omega \mu_{0}\sigma_{m}}}=\sqrt{\frac{1}{\pi f\mu_{0}\sigma_{0}}}$$
where $\sigma_{m}$ is the conductivity material of the movable membrane and f is the operating frequency of the developed capacitive RF MEMS switches.

Based on the calculations carried out, it follows that RF MEMS switches with a low resistance $R_{m}$ of the movable membrane demonstrate a sharper and deeper resonance at the operating frequency of the RF signal, as well as a lower resistance $R_{m}$ of the movable membrane, 0.01–0.1 Ω, which provides a higher value of isolation in the closed state of the switch. For example, the resistance of a movable membrane $R_{m}$ of 10 Ω for the same resonant frequency of the RF signal provides an isolation value of 10 dB. Similarly, the lower resistance of the movable membrane $R_{m}$ demonstrates lower values of return loss in the closed state. Thus, the resistance of the movable membrane $R_{m}$ of the developed capacitive RF MEMS switches is 0.045 Ω and 0.033 Ω, respectively, and the total resistance $R_{b}$ in the closed state of the switches is 0.7 Ω and 0.62 Ω, respectively.

In addition, the inductance of the movable membrane $L_{b}$ affects the EM parameters when the capacitive RF MEMS switch is in the closed position, whereas in the open state the inductance $L_{b}$ has little effect. At higher inductance values of the movable membrane $L_{b}$, resonance occurs at a lower frequency of the RF signal, whereas at lower inductance values $L_{b}$, resonance occurs at higher frequencies. Therefore, if a higher isolation value is required at lower frequencies, then in this case a high inductance value of the movable membrane is preferable, while the effect of the inductance value $L_{b}$ on the value of the return loss in the closed position of the switch is nominal.

In the developed designs of movable membranes, in order to reduce the damping effect of the compressible metal layer and increase the switching speed, through holes of various geometries are provided: for switch (A)—circle, for switch (B)—square. The area of the holes is 18.36% and 12.36%, respectively, of the total area of the movable membranes. The pattern of holes is characterized by the following geometric parameters: for the movable membrane of switch (A) step $\Delta_{1}$ = 3μm—the distance between adjacent holes and step $\Delta_{1}$ = 7μm—the distance from the nearest hole to the edge; for the movable membrane of switch (B) step $\Delta_{1}$ = 6μm—the distance between adjacent holes and step $\Delta_{1}$ = 8μm—the distance from the nearest hole to the edge. The efficiency can be determined using the following ratio: μ = l/p, where l is the distance from the edge of one hole to the edge of the neighboring hole; p is the distance from the center of the hole to the center of the neighboring hole. The efficiency of the hole pattern is $\mu_{1}$ = 0.43 and $\mu_{1}$ = 0.6, respectively.

These holes are also designed to reduce residual stress in movable membranes and reduce Young’s modulus E. In addition, the holes also lead to a decrease in the mass of the movable membranes, which, in turn, leads to a higher mechanical resonance frequency.

Figure 13 shows a schematic representation of the movement of one of the parts of the movable membrane and the distribution of the air flow under it. The movable membrane is located above a fixed plane—the fixed down actuation electrode with an initial height of the air gap $g_{0}$ and movement in the direction of the Z axis. As a result of moving the movable membrane into the pressure under it, it will melt due to the small air gap. Consequently, the available air will be displaced laterally and upwards through holes, as shown in Figure 13. The radius of the holes is equal to $r_{0}$ in increments of $\xi_{0}$. The arrows indicate the direction of the compressed air flow. When considering the boundary conditions the damping pressure created under the movable membrane can be determined using Equation (Equation 22):(22)$$P\left(x,\;t\right)=\frac{6\mu }{g_{0}^{3}}\frac{dg_{0}}{\mathrm{dt}}x^{2}$$

The value of $P\left(x,t\right)$ is positive in the case where the air under the movable membrane is compressed, $dg_{0}/\mathrm{dt}$ < 0, and vice versa. The damping force F on the movable membrane is determined by Equation (Equation 23):(23)$$F=-\frac{\mu b^{3}l}{g_{0}^{3}}\frac{dg_{0}}{\mathrm{dt}}$$

The speed of movement of the oscillating movable membrane under the influence of electrostatic forces is uneven, and the distribution mainly depends on the vibration mode. Since the air flow caused by vibration mainly has lateral directions due to the small air gap $g_{0}$, the damping force coefficient is determined using Equation (Equation 24):(24)$$\overset{ˇ}{c}=\frac{\mu b^{3}l}{t^{3}}$$

According to the equation of bending moment and moment of inertia $M\left(X\right)$ = −$EI\dot{w}\left(x\right)$, one obtains the equation $\mathrm{EI}\left(\partial^{4}/\partial x^{4}\right)$+$\rho A\left(\partial^{2}/\partial t^{2}\right)$ = $q\left(x\right)$ and $\overset{ˇ}{F}$ = $\left(\mu b^{3}/g_{0}^{3}\dot{w}\left(x,t\right)\right)$, where $\dot{w}\left(x,t\right)$ is the speed of movement of the sector of the movable membrane. The differential Equation (Equation 25) for the damping force of a compressed layer with air under forced vibration of a movable membrane will have the following form:(25)$$\rho \mathrm{bd}\ddot{w}\left(x,\;t\right)+\overset{ˇ}{c}\dot{w}\left(x,\;t\right)+\mathrm{EI}\frac{\partial^{4}w\left(x,\;t\right)}{\partial x^{4}}$$
where ρ is the mass density; I is the moment of inertia; E is the Young’s modulus of the material used for the movable membrane; and $\overset{ˇ}{c}$ is the damping coefficient per unit length of the movable membrane.

The ξ for the n vibration mode is determined by Equation (Equation 26):(26)$$\xi_{n}=\frac{\ddot{c}}{2\rho \mathrm{bd}\omega_{n}}=\frac{\mu b^{2}}{2\rho dg_{0}^{3}\omega_{n}}$$

The quality factor of the movable membrane (for a small damping coefficient) is determined using Equation (Equation 27):(27)$$Q=\frac{\rho dg_{0}^{3}\omega_{n}}{\mu b^{2}}$$

The quality coefficient of the developed movable membrane designs is Q = 0.7. It is important to use through holes in the movable membrane, especially for small air gaps $g_{0}$. At very low air pressures, both the damping coefficient b and the removal of air from under the movable membrane are limited by scattering at the anchor areas and the boundaries of the CPW. At the same time, the permissible value of the quality coefficient is Q > 3; if this condition is met, the quality factor will not have a strong effect on the switching speed but it will negatively affect the opening time of the switch.

Another parameter that should be taken into account is the dimensionless squeeze number σ determined using Equation (Equation 28):(28)$$\sigma =\frac{12\mu \omega l^{2}}{P_{a}g_{0}^{2}}$$
where $P_{a}$ is the packaging pressure.

### 3.2. Design of the Actuation Electrodes

To implement an electrostatic control mechanism in the developed designs of capacitive RF MEMS switches, a control voltage is applied between a movable electrode—the movable membrane connected to the grounding lines of the CPW using anchor areas and fixed down actuation electrodes.

Figure 14 shows a schematic representation of the structures of the fixed down actuation electrodes of the developed capacitive RF MEMS switches and their basic geometric dimensions, as well as a model of force distribution; ξ is the force applied in the vertical direction Z during electrostatic actuation from $x_{1}$ to $x_{2}$. As a result, the force is distributed over 2/3 of the area in the movable membrane—the left and right parts, respectively, for switch (A) and switch (B).

The force distribution can be estimated using Equation (Equation 29):(29)$$y=\frac{2}{\mathrm{EI}}\int_{x_{1}}^{x_{2}}\frac{\xi }{48}\left(l^{3}-6l^{2}a+9la^{2}\right)$$
where ξ is the load per unit length l; I is the moment of inertia; and E is the Young’s modulus of the material. Integrating Equation (Equation 23) from $x_{1}$ to $x_{2}$ will give an approximate force distribution.

The moment of inertia I is given by Equation (Equation 30):(30)$$I=\frac{wt^{3}}{12}$$
where w and *t* are the width and thickness of the movable membrane, respectively.

The load per unit length can be calculated using Equation (Equation 31):(31)$$P=\xi 2\left(x_{1}-x_{2}\right)$$

In this case, the movable membrane is affected by the force of electrostatic interaction $F_{e}$, which is balanced by the elastic force—restoring force $F_{r}$, depending on the coefficient of elasticity of the elastic suspension elements. The balance of forces $F_{e}=F_{r}$ exists as long as the elastic force, which is a linear function, can compensate for the growth of the electrostatic interaction force $F_{e}$, which varies according to the quadratic law. At some point, the growth of the elastic force $F_{r}$ cannot compensate for the growth of the electrostatic interaction force $F_{e}$, and the movable electrode falls on the fixed down actuation electrodes. The electrostatic force $F_{e}$ applied between the movable membrane and the fixed down actuation electrodes can be calculated using Equation (Equation 32):(32)$$F_{e}=\frac{1}{2}V^{2}\frac{\mathrm{dC}(g_{0})}{dg_{0}}=\frac{1}{2}\frac{\varepsilon_{0}\mathrm{Ww}V^{2}}{g_{0}}$$
where V is the applied control voltage to the fixed down actuation electrodes.

The value of the threshold voltage can be determined using the balance of electrostatic $F_{e}$ and restoring $F_{r}$ forces using Equation (Equation 33):(33)$$\begin{array}{c} F_{e}=F_{r}=K_{z}\left(g-g_{0}\right);\;\frac{1}{2}\frac{\varepsilon_{0}\mathrm{Ww}V^{2}}{g}=K_{z}\left(g-g_{0}\right);\;V_{\mathrm{th}}=\sqrt{\frac{2K_{z}g^{2}\left(g-g_{0}\right)}{\varepsilon_{0}\mathrm{Ww}}} \end{array}$$
where $K_{z}$ is the total component of the stiffness of the movable membrane; *W* is the width of the fixed down actuation electrode; w is the width of the corresponding part of the movable electrode; and $g_{0}$ is the initial air gap between the movable membrane and the fixed down actuation electrodes at zero control voltage.

At $2/3g_{0}$, the electrostatic force $F_{e}$ becomes significantly greater than the restoring force $F_{r}$. Hence, the value of the pull-down voltage can be determined using Equation (Equation 34):(34)$$V_{p}=V\left(\frac{2g_{0}}{3}\right)=\sqrt{\frac{8K_{z}}{27\varepsilon_{0}\mathrm{Ww}g_{0}^{3}}}$$

Figure 14 also shows the contact pads for connection to the corresponding contacts in the RF package using micro-welding with Au wire and contact paths to the fixed down actuation electrodes, on the surface of which a thin layer of $\mathrm{Si}O_{2}$ is applied to electrically isolate the control voltage from the RF signal and prevent a short circuit. The gaps in the grounding lines of the CPW under the contact pads are connected by a metal conductive layer of Cu to equalize the electrical potential on the grounding lines of the CPW.

Figure 15 show the results of modeling the scattering parameters, as well as the results of the associated simulation of the transient thermal modeling during the passage of a 1W RF signal for the developed CPW (A) and CPW(B), including fixed down actuation electrodes, contact pads, contact lines to study the effect on scattering parameters and temperature distribution in the structure.

From the data presented in Figure 15, it can be seen that the developed fixed down actuation electrodes, contact pads and contact lines included in the design of CPW (A) and CPW (B) do not significantly affect the scattering parameters and temperature rise when switching an RF signal with a power of 1W. The effect on insertion loss $S_{11}$ for CPW (A) is an increase of −0.01 dB, for CPW (B)—a decrease of −0.01 dB; the reflection loss $S_{12}$ for CPW (A)—a decrease of −5.53 dB, for CPW (B)—a decrease of −14.96 dB, which indicates the absence of influence on the coordination of RF input and RF output, and the absence of influence on the value of the VSWR; the increase in maximum temperature for CPW (A)—by $5.19{\;}^{\circ }C$ and for CPW (B)—by $0.99{\;}^{\circ }C$ is partially due to a delayed heat sink due to a small air gap in the location of the fixed down actuation electrodes relative to the RF signal line of the CPW.

### 3.3. Design of the Elastic Suspension Elements

Elastic suspension elements, made in the form of a zig-zag shape, make it possible to reduce the stiffness coefficient of the movable membrane of the capacitive RF MEMS switch by a necessary and sufficient amount, reduce the mechanical peak tension by reducing stiffness in an undesirable direction, and reduce the amount of axial mechanical stress while compromising the high switching speed and the required design power of the switched RF signal.

Figure 16 schematically shows the developed zig-zag elastic suspension used to provide a small amount of residual thermal and mechanical stress in the developed designs of switch (A) and switch (B). Each zig-zag elastic suspension consists of five successively connected elastic beams with two-fold symmetry, forming the shape of a meander, as well as one connecting elastic beam. Each movable membrane of the developed capacitive RF MEMS switches is suspended using four zig-zag elastic suspensions fixed to the corresponding anchor areas, which are located on the grounding lines of the CPW, as shown in Figure 5.

To determine the value of the efficiency of the stiffness coefficient $K_{\mathrm{eff}}$, it is enough only to analyze one elastic suspension and the resulting effective stiffness coefficient will be equal to $K_{\mathrm{eff}}$ = K/4 of the stiffness coefficient K of one zig-zag elastic suspension.

The effective stiffness coefficient of the presented zig-zag elastic suspension is determined using Equation (Equation 35):(35)$$K=K_{z}+K_{b}$$
where $K_{z}$ is the stiffness coefficient of the elastic suspension in the form of a zig-zag and $K_{b}$ the stiffness coefficient of the connecting elastic suspension in the form of a cantilever beam, respectively.

The stiffness coefficient $K_{z}$ of a zig-zag elastic suspension is determined by Equation (Equation 36) [24]:(36)$$\begin{array}{c} K_{z}=\frac{48GJ}{l_{a}^{2}\left(\frac{\mathrm{GJ}}{EI_{x}}l_{a}+l_{b}\right)n^{3}},\;n\gg \frac{3l_{b}}{\frac{\mathrm{GJ}}{EI_{x}}l_{a}+l_{b}} \end{array}$$
where E is the Young’s modulus; G is the shear modulus, defined by the following expression: G = E/(2(1+ν)); ν is the Poisson’s ratio; J is the torsion constant, defined by the following expression: J = $0.413I_{p}$; $I_{p}$ is the polar moment of inertia, defined by the expression: $I_{p}$ = $I_{x}+I_{z}$; $I_{x}$ is the moment of inertia along the axis X, defined by the expression: $I_{x}$ = $wt^{2}/12$; $I_{z}$ is the moment of inertia along the axis Z, determined by the expression: $I_{z}$ = $tw^{3}/12$; and n is the number of meanders.

The stiffness coefficient $K_{b}$ of an elastic suspension in the form of an elastic beam is determined using Equation (Equation 37):(37)$$K_{b}=\mathrm{Ew}{\frac{t}{l}}^{3}$$

The switching time can be analytically estimated using Equation (Equation 38):(38)$$t_{s}=3.67\frac{V_{p}}{V_{s}\omega_{0}}$$
where $V_{s}$ is the switching voltage equal to 1.2–1.4$V_{p}$ and $\omega_{0}$ is the mechanical resonant frequency of the switch determined using the following Equation (Equation 39):(39)$$\omega_{0}=\frac{1}{2\pi }\sqrt{\frac{K_{\mathrm{eff}}}{M_{\mathrm{eff}}}}$$
where $M_{\mathrm{eff}}$ = $0.35\left(\mathrm{lwt}\right)\rho$ is the effective mass of the movable switch membrane.

Table 7 shows the various geometric parameters of the zig-zag elastic suspension, as well as the analyzed electromechanical parameters of switch (A) and switch (B).

Figure 17 and Figure 18 present the results of modeling the electromechanical parameters of the movable structures of the developed capacitive RF MEMS switches.

Based on the data presented in Figure 17 and Figure 18, and the electromechanical modeling carried out, the developed designs of zig-zag elastic suspensions and movable membranes of the developed capacitive RF MEMS switches are characterized by a small effective coefficient of stiffness, and at the same time a small value of the pull-down voltage and switching time.

### 3.4. Design of the MIM Capacitor

Figure 19 shows a schematic representation of an additional fixed MIM capacitor for the developed capacitive RF MEMS switches and their geometric dimensions.

In the case of a typical design of the capacitive RF MEMS switch, the capacitance $C_{\mathrm{on}}$ is in the up-position of the movable membrane (open state) and the capacitance is in the down-position $C_{\mathrm{off}}$ (closed state) accordingly; it is determined using Equation (Equation 40):(40)$$C_{\mathrm{on}}=\varepsilon_{0}A_{\mathrm{up}}{\left(g_{0}+\frac{t_{d}}{\varepsilon_{r}}\right)}^{-1};\;C_{\mathrm{off}}=\varepsilon_{0}A_{d}\frac{t_{d}}{\varepsilon_{r}}$$
where $\varepsilon_{0}$ is the permittivity of the air space; $g_{0}$ is the initial value of the air gap; $\varepsilon_{r}$ is the relative permittivity of the dielectric layer; $t_{d}$ is the thickness of the dielectric layer; and $A_{\mathrm{up}}$ and $A_{d}$ are the overlap area of the movable membrane and the fixed down actuation electrodes in the up- and down-positions of the movable membrane, respectively.

Therefore, if we neglect the edge effects, the capacitance ratio in the down-position to the capacitance in the up-position of the movable membrane of a typical capacitive RF MEMS switch can be expressed using Equation (Equation 41):(41)$$C_{r}=\frac{C\mathrm{off}}{C\mathrm{on}}=\frac{A_{d}}{A_{\mathrm{up}}}\left(1+g_{0}\frac{\varepsilon_{r}}{t_{d}}\right)$$

It follows that the value of the capacitance ratio $C_{r}$ is limited by three factors: (1) the relative permittivity of the dielectric layer $\varepsilon_{r}$; (2) the thickness of the dielectric $t_{d}$; and (3) the initial value of the air gap $g_{0}$. These limiting factors are not solved easily. Firstly, the value of the dielectric permittivity $\varepsilon_{r}$ is determined precisely only at the stage of working out the manufacturing process route; secondly, the problem of the charge of the dielectric layer is significant when a thin dielectric layer is used; thirdly, the large value of the initial air gap $g_{0}$ leads to high values of the control voltage.

As shown in Figure 19, in the developed design of an additional fixed MIM capacitor the parameter δ was determined, which is equal to $A_{\mathrm{MIM}}$/$A_{\mathrm{MAM}}$. If the marginal edge effects of the capacitance value in the up- and down-positions of the movable membrane are neglected, the value of the capacitance ratio $C_{r}$ can be expressed using Equation (Equation 42):(42)$$\begin{array}{c} C_{r}=\frac{C_{d}}{C_{\mathrm{up}}}=\frac{C_{\mathrm{MIM}}}{C_{\mathrm{MIM}}\|C_{\mathrm{MAM}}}=\frac{C_{\mathrm{MIM}}}{C_{\mathrm{MAM}}}+1=\frac{\varepsilon_{r}g_{0}}{t_{d}}\frac{A_{\mathrm{MIM}}}{A_{\mathrm{MAM}}}+1=\frac{\varepsilon_{r}g_{0}}{t_{d}}\delta +1\\ C_{\mathrm{MIM}}=\frac{\varepsilon_{0}\varepsilon_{r}}{t_{d}}A_{\mathrm{MIM}};\;C_{\mathrm{MAM}}=\frac{\varepsilon_{0}}{g_{0}}A_{\mathrm{MAM}};\;g_{0}=\frac{\left(\frac{C_{r}-1}{\delta }\right)t_{d}}{\varepsilon_{r}} \end{array}$$

In this case, Equation (Equation 34) for determining the value of the pull-down voltage can be represented as the following Equation (Equation 43):(43)$$V_{p}=\sqrt{\frac{8k{\left(\frac{C_{r}-1}{\delta }\right)}^{3}t_{d}^{3}}{27\varepsilon_{0}A_{p}\varepsilon_{r}^{3}}}$$

Table 8 presents the results of the analytical calculation of the MIM capacitors of the developed capacitive RF MEMS switches.

Figure 20 shows the results of modeling the scattering parameters, as well as the results of the associated simulation of the transient thermal modeling during the passage of a 1W RF signal for the developed CPW (A) and CPW(B), including fixed down actuation electrodes, contact pads, contact lines and additional fixed down MIM capacitor, to study the effect on scattering parameters and temperature distribution in the structure.

From the data presented it can be seen that the developed fixed down actuation electrodes, contact pads, contact lines and MIM capacitor included in the design of CPW (A) and CPW (B) do not significantly affect the scattering parameters and temperature rise when switching an RF signal with a power of 1W. The effect on insertion loss $S_{11}$ for CPW (A) is not changed; for CPW (B)—a decrease of −0.01 dB; the reflection loss $S_{12}$ for CPW (A)—an increase of −2.2 dB, for CPW (B)—a decrease of −8.9 dB, which indicates the absence of influence on the coordination of RF input and RF output, and the absence of influence on the value of the SWR; the increase in maximum temperature for CPW (A)—by $0.56{\;}^{\circ }C$, for CPW (B)—a decrease of $2.41{\;}^{\circ }C$; this is due to the increased heat dissipation of the induced heat through the additional metal lining of the MIM capacitor.

Figure 21 shows the results of modeling the scattering parameters, as well as the results of the associated simulation of the transient thermal modeling during the passage of a 1W RF signal for the developed capacitive RF MEMS switch (A) in the open- and closed state, respectively.

Figure 22 show the results of modeling the scattering parameters, as well as the results of the associated simulation of the transient thermal modeling during the passage of a 1W RF signal for the developed capacitive RF MEMS switch (B) in the open- and closed state, respectively.

From the data presented in Figure 21 it can be seen that the developed RF MEMS switch (A) is characterized by good EM parameters, as well as the possibility of switching RF signals with a power of 1W or more. According to the simulation of EM parameters, the scattering parameters in the open state of the switch should be: insertion loss $S_{11}$ not more than −0.07 dB and reflection loss $S_{12}$ not more than −41.17 dB. When an RF signal with a power of 1W is passed through the switch CPW, the maximum temperature is $86.7{\;}^{\circ }C$, distributed over zig-zag elastic suspensions. In the closed state of the switch, the return loss of $S_{11}$ is not more than −0.16 dB and the isolation value $S_{21}$ is not less than −42.2 dB. The central resonant frequency of this switch is 3.6GHz and the effective frequency range covers the S-band. When switching an RF signal with a power of 1W, the maximum temperature is $293.17{\;}^{\circ }C$, also distributed over zig-zag elastic suspensions. This temperature is not critical, i.e., the softening or melting temperature is $660.3{\;}^{\circ }C$ for the material used for the movable membrane and zig-zag suspensions and, for this reason, there are no induced arbitrary displacements of the movable elements of the switch design or their irreversible deformation.

From the data presented in Figure 22 it can be seen that the developed RF MEMS switch (B) is characterized by good EM parameters, as well as the possibility of switching RF signals with a power of 1W or more.

According to the simulation of EM parameters, the scattering parameters in the open state of the switch should be: insertion loss $S_{11}$ not more than −0.16 dB and reflection loss $S_{12}$ not more than −30.8 dB. When an RF signal with a power of 1W is passed through the switch CPW, the maximum temperature is $144.4{\;}^{\circ }C$, distributed over zig-zag elastic suspensions. In the closed state of the switch, the return loss of $S_{11}$ is not more than −0.2 dB and the isolation value $S_{21}$ is not less than −54.9 dB. The central resonant frequency of this switch is 3.4GHz, and the effective frequency range covers the C-, X- and Ku-bands, in which the isolation value is at least −30 dB. When switching an RF signal with a power of 1W, the maximum temperature is $386.5{\;}^{\circ }C$, also distributed over zig-zag elastic suspensions. This temperature is not critical, i.e., the softening or melting temperature is $660.3{\;}^{\circ }C$ for the material used for the movable membrane and zig-zag suspensions, and for this reason there are no induced arbitrary displacements of the movable elements of the switch design or their irreversible deformation.

## 4. Manufacturing Process and Experimental Research

Figure 23 shows the sequence of application of technological layers in the manufacture of experimental samples of RF MEMS switch (A) and RF MEMS switch (B).

Figure 24 shows photographs obtained using an electron microscope in the process of studying and improving the manufacturing process route.

Figure 25 shows photographs of manufactured experimental samples of RF MEMS switches in a coaxial RF package for experimental studies.

For the purposes of conducting experimental studies of manufactured RF MEMS switches, namely, providing external connection to electrical terminals, for protection from external electric and magnetic fields, and sealing, a package method was chosen based on a package of separated RF MEMS switches on the circuit board by fixing them in a specialized package for RF devices. This package is designed to work in electrical circuits transmitting RF signals of high frequency (up to 10 GHz) with matching RF input and RF output in 50 Ω. RF input and RF output of this RF package are coaxial connectors that minimize losses at the connection points.

Figure 26 shows the results of experimental studies of electromagnetic parameters of manufactured experimental samples of RF MEMS switches in a specialized RF package.

The parameters of RF signal transmission in the frequency range from 0 Hz to 10 GHz of the manufactured experimental samples of RF MEMS switches were measured using a vector analyzer of electrical circuits (Rohde & Schwarz ZVB-20) and a linear control voltage supply source (GW Instek-73303S).

From the data presented in Figure 26 it can be seen that the manufactured experimental sample of RF MEMS switch (A) is characterized by good EM parameters, as well as the possibility of switching RF signals with a power of 1W or more. According to the simulation of EM parameters in the open state of the switch should be: insertion loss $S_{11}$ they are not more than −0.69 dB and the reflection loss of $S_{12}$ are not more than −28.25 dB. In the closed state of the switch the isolation value $S_{21}$ is −54.77 dB. The central resonant frequency of this switch is 3.6GHz and the effective frequency range covers the S-band.

From the data presented in Figure 26 it can be seen that the manufactured experimental sample of RF MEMS switch (B) is characterized by good EM parameters, as well as the possibility of switching RF signals with a power of 1W or more. According to the simulation EM parameters in the open state of the switch should be: insertion loss $S_{11}$ not more than −0.66 dB and reflection loss $S_{12}$ not more than −20.66 dB. In the closed state of the switch the isolation value $S_{21}$ is −52.13 dB. The central resonant frequency of this switch is 3.4GHz and the effective frequency range covers the C-, X- and Ku-bands, in which the isolation value is at least −30 dB.

The differences between experimental data and theoretical data, namely, the magnitude of insertion loss in the open state of the manufactured experimental samples of RF MEMS switches, are explained by the increasing insertion loss on the conductor wire connections and the adapter board intended for installation in the RF package.

A separate stage of experimental research was the study of the process of obtaining ${\mathrm{TiO}}_{2}$ thin dielectric films by reactive magnetron sputtering on glass and sapphire substrates, as shown in Figure 27, in order to achieve the required dielectric characteristics. The studies were carried out by spraying capacitor structures with metal–dielectric–metal plates. The material of the lower and upper metal lining was Al, Cu, Mo, Ti. The thickness of the metal films was 250 nm. The thickness of the ${\mathrm{TiO}}_{2}$ dielectric layer was 200 nm. The ${\mathrm{TiO}}_{2}$ target was a washer with a diameter of 100 mm and a thickness of 5 mm, which was placed above the water-cooled magnetron table at a distance of 5 cm from the fixed and pre-cleaned substrates. To ensure the uniformity of the films in thickness, the table rotated at a speed of five revolutions per minute. The temperature of the substrates was controlled by a system of thermocouples in a vacuum chamber. To remove unwanted impurities in the vacuum chamber, short-term etching of the target surface with bombarding argon ions was performed. The thickness of the deposited films was monitored using an interferometer. Then the volt–farad characteristics of the obtained capacitor structures were measured and the dielectric permittivity of the resulting dielectric film was determined by analytical calculation.

## 5. Conclusions

Table 9 shows a comparative analysis of the designed RF MEMS switches with similar RF MEMS switches suitable for applications in 5G mobile network devices.

In this article, based on the developed methodology for designing high-performance capacitive RF MEMS switches for certain applications and devices based on the operating resonant frequency or frequency range, the stages were researched of designing two designs of high-performance RF single-pole single-throw (SPST) MEMS switches with a hybrid contact type, as well as using a dielectric layer with a high dielectric permittivity $\mathrm{Ti}O_{2}$ (high-k). The RF MEMS switches are designed to operate at a central resonant frequency of 3.6 GHz and 3.4 GHz, respectively, as well as to work both in mobile communication devices and in the design of the architecture of mobile networks of the 5th generation—5G, in particular in arrays of integrated antennas and RFFE. For the purposes of testing the developed design methodology, the manufacture and study of two designed structures of the developed high-performance RF MEMS switches with a hybrid contact type based on surface processing technology was carried out. According to the results of the study, the manufactured experimental samples of two high-performance RF MEMS switches with a hybrid contact type are characterized by good electromechanical and electromagnetic parameters, and high linearity, as well as the possibility of switching RF signals with a power of more than 1 W. In this case, the value of the control voltage is 3.5 and 5 V, and the switching time is 6.35 μs and 6.5 μs, respectively. For the first manufactured experimental sample of the RF MEMS switch in the open state the insertion loss is no more than −0.69 dB and the reflection loss is −28.35 dB, and in the closed state the isolation value is at least −54.77 dB at a central resonant frequency of 3.6 GHz and the effective frequency range covers the S-range, in which the isolation value is at least −30 dB. For the second manufactured experimental sample of the RF MEMS switch in the open state the value of the insertion loss is no more than −0.66 dB and the reflection loss is −20.66 dB, and in the closed state the isolation value is not less than −52.13 dB at the central resonant frequency of 3.4 GHz and the effective frequency range covers the C-, X- and Ku-ranges in which the isolation value is not less than −30 dB. The differences between experimental data and theoretical data, namely, the magnitude of insertion loss in the open state of the manufactured experimental samples of RF MEMS switches, are explained by the increasing insertion loss on the conductor wire connections and the adapter board intended for installation in the RF package. Both manufactured experimental samples of the RF MEMS switches are characterized by high linearity, as well as a small value of contact resistance in the closed state of 0.7 Ω and 0.62 Ω, respectively. The results of an experimental study of manufactured experimental samples of high-performance RF MEMS switches with a hybrid contact type show good agreement with the theoretical results obtained during the design using the developed design methodology, which demonstrates its effectiveness as a suitable tool for engineers when designing high-performance capacitive RF MEMS switches for certain applications and devices.

The results obtained can be used as a foundation for designing complex RF MEMS networks, such as multi-position attenuators (for example, 5–8 bits), the development of which would also reduce hardware redundancy in RFFE designed for 5G mobile networks. In addition, more RF signal generation functions can be combined in the same RF MEMS device, for example, attenuation and phase shift, which makes such a technical solution even more attractive for 5G applications. In general, the engineering and development of an element component base based on passive RF MEMS elements, such as RF antenna switches, intermediate frequency filters, LC-filters and resonators, will lead to the replacement of traditional active semiconductor analogs, while increasing the performance of 5G mobile networks and simultaneously causing a rethinking of the architecture of RF transceivers.

## Figures and Tables

**Figure 1 micromachines-14-00477-f001:**
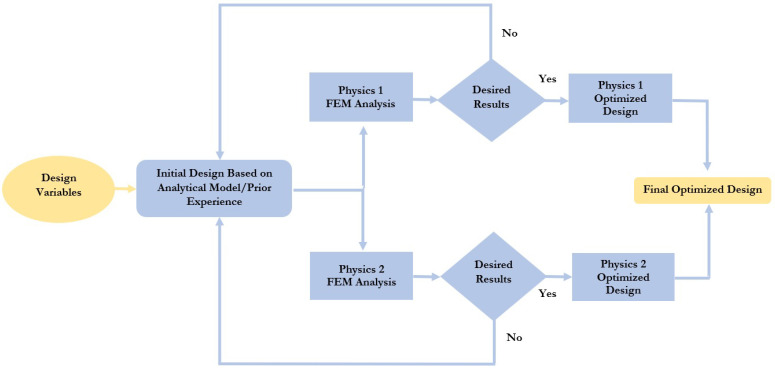
Traditional design optimization approach for MEMS devices.

**Figure 2 micromachines-14-00477-f002:**
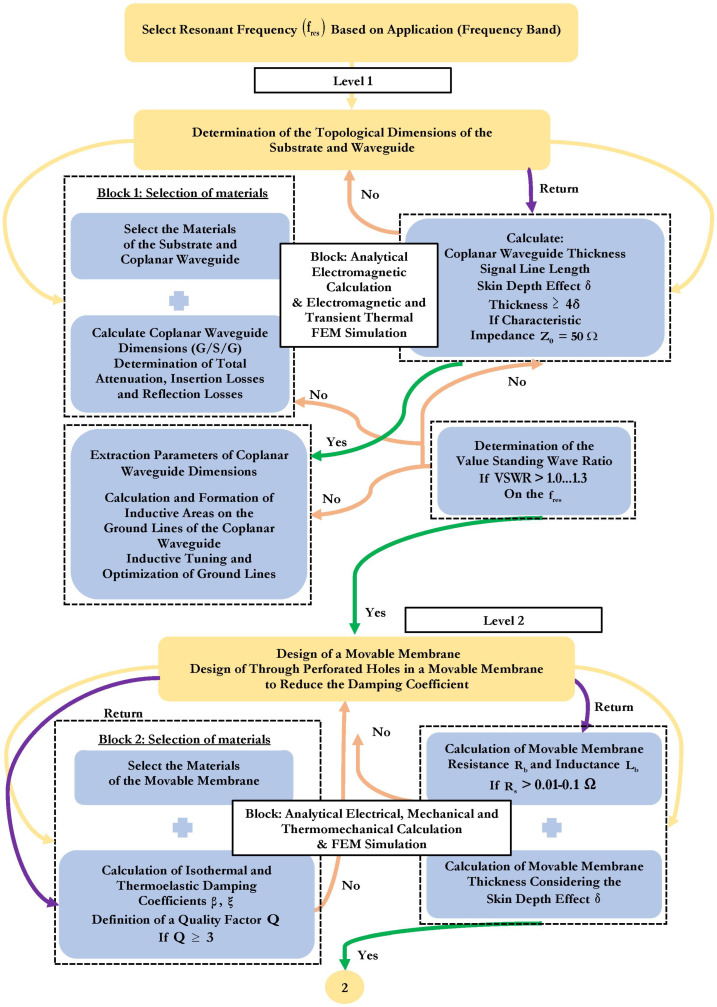
The proposed methodology for optimizing capacitive RF MEMS switches.

**Figure 3 micromachines-14-00477-f003:**
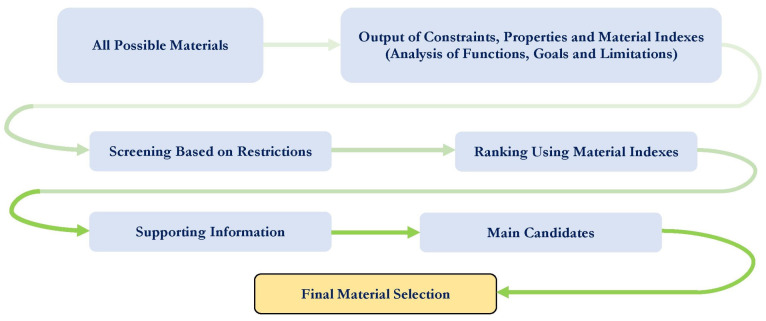
Methodology of material selection.

**Figure 4 micromachines-14-00477-f004:**
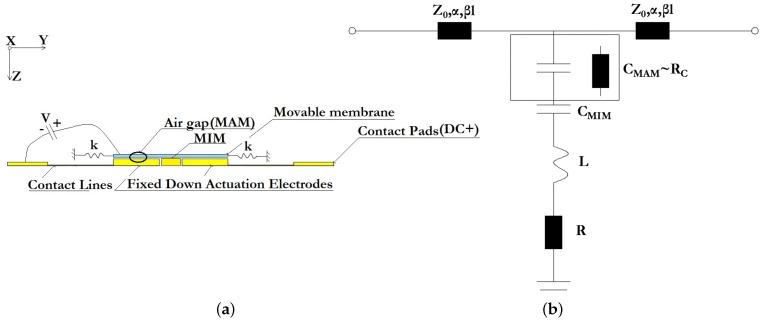
Schematic view of the developed capacitive RF MEMS switches: (**a**) the one-dimensional model of the displacement of the movable membrane during electrostatic activation; (**b**) the equivalent electrical model.

**Figure 5 micromachines-14-00477-f005:**
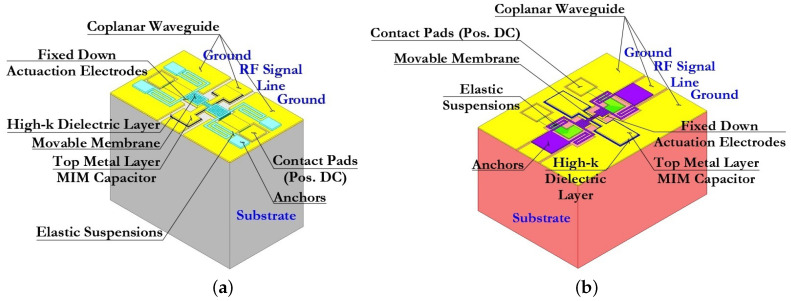
The isometric 3D topology of the developed capacitive RF MEMS switches: (**a**) switch (A); (**b**) switch (B).

**Figure 6 micromachines-14-00477-f006:**
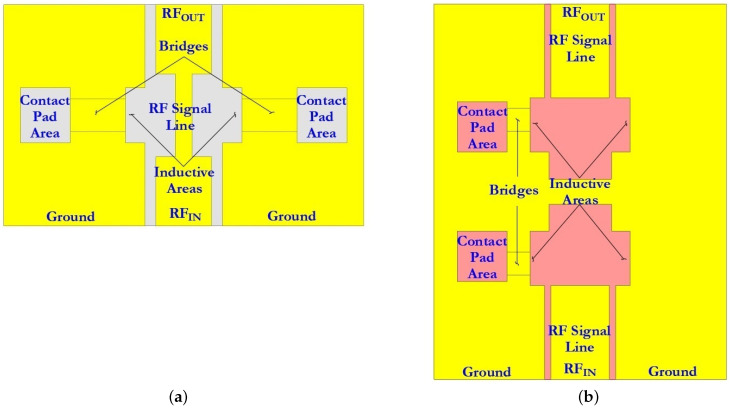
The CPW designs: (**a**) CPW (A) is RF MEMS switch (A); (**b**) CPW (B) is RF MEMS switch (B).

**Figure 7 micromachines-14-00477-f007:**
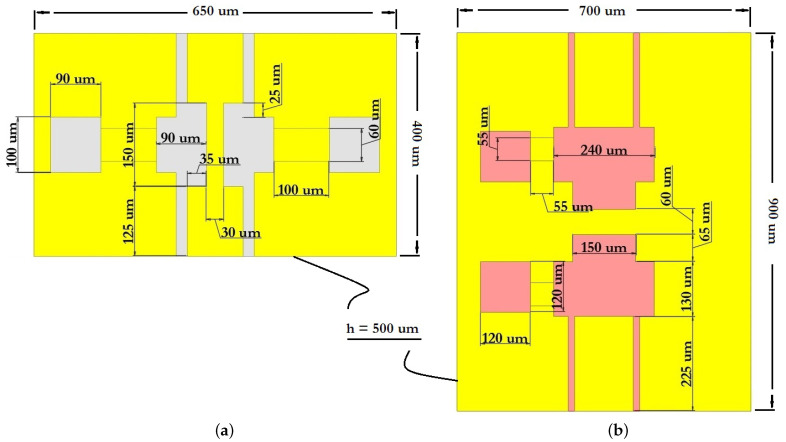
Schematic view: (**a**) CPW (A) is RF MEMS switch (A); (**b**) CPW (B) is RF MEMS switch (B).

**Figure 8 micromachines-14-00477-f008:**
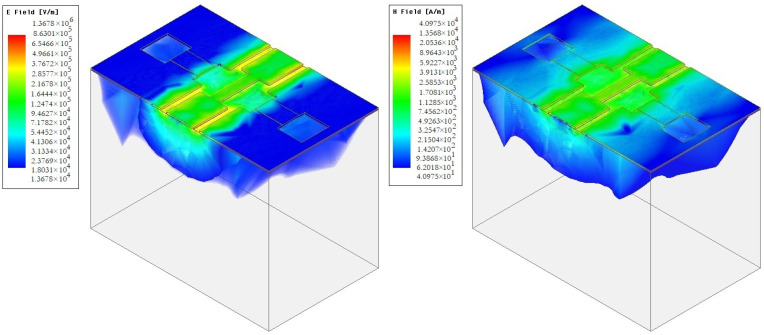
The distribution of E and H fields in the developed design of CPW (A).

**Figure 9 micromachines-14-00477-f009:**
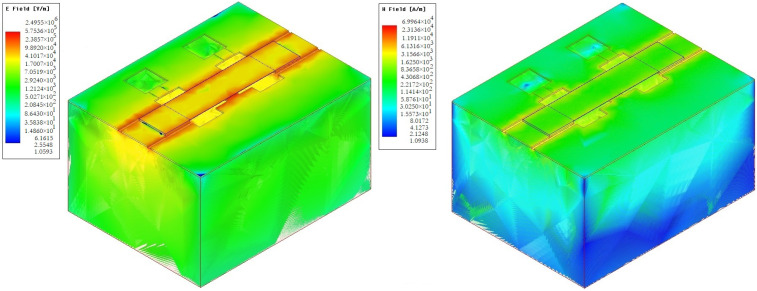
The distribution of E and H fields in the developed design of CPW (B).

**Figure 10 micromachines-14-00477-f010:**
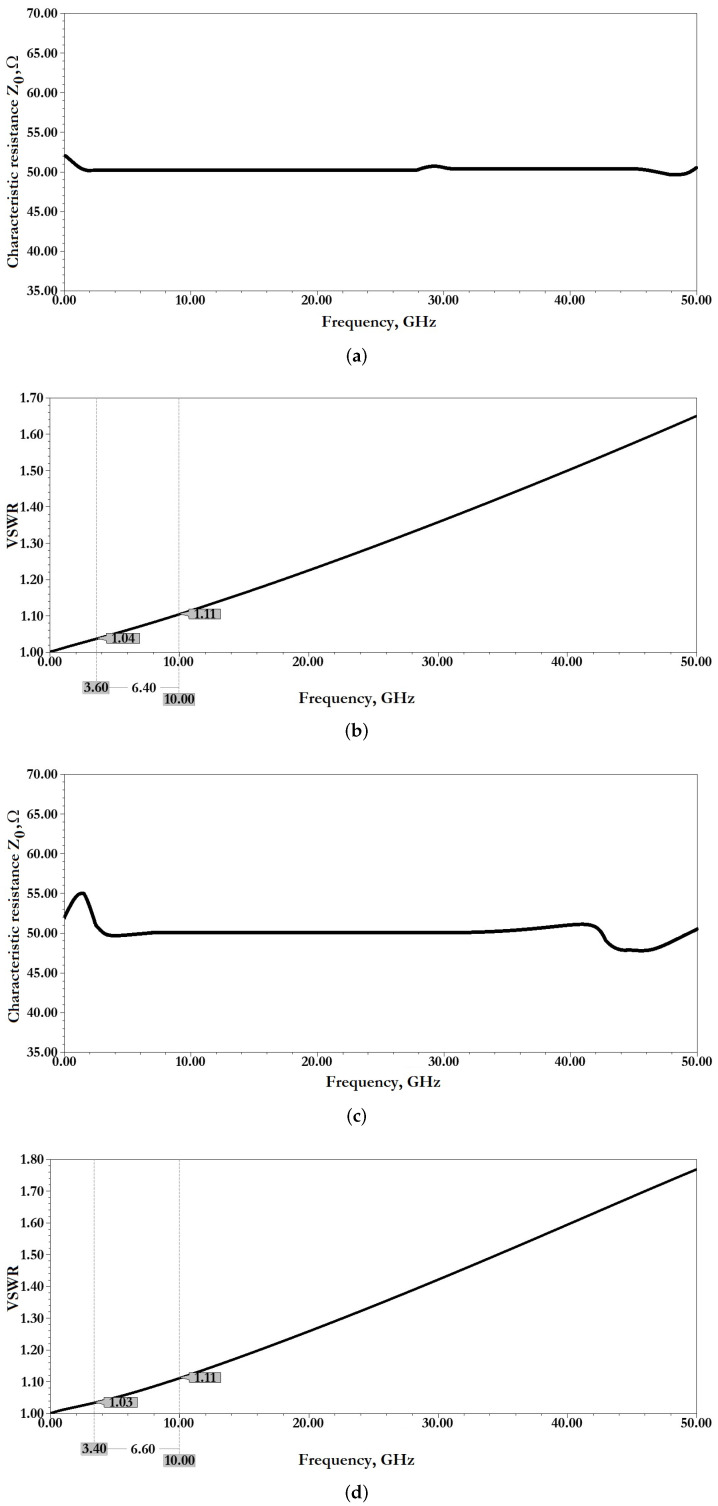
Distribution graph of CPW (A) and CPW (B) from the frequency RF signal: (**a**,**b**) the characteristic resistance $Z_{0}$ and the VSWR of CPW (A); (**c**,**d**) the characteristic resistance $Z_{0}$ and the VSWR of CPW (B).

**Figure 11 micromachines-14-00477-f011:**
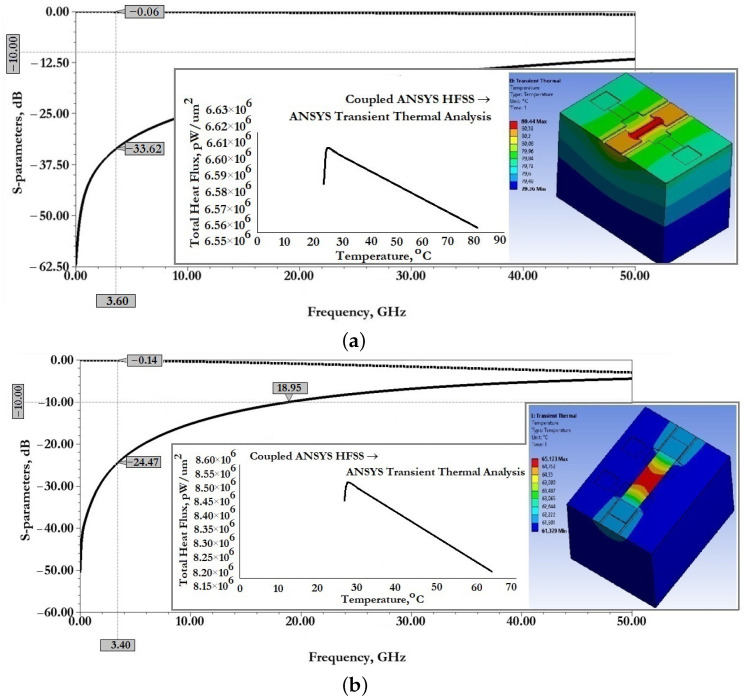
The results of EM and transient thermal modeling: (**a**) CPW (A); (**b**) CPW (B).

**Figure 12 micromachines-14-00477-f012:**
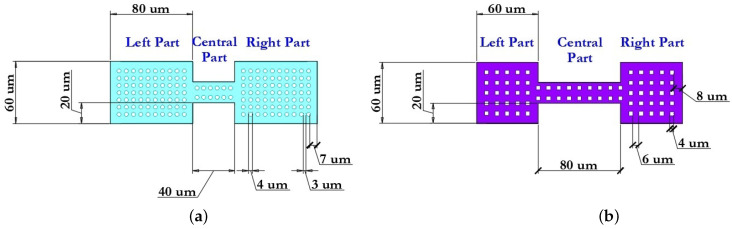
Schematic view of the designs of the movable membranes: (**a**) switch (A); (**b**) switch (B).

**Figure 13 micromachines-14-00477-f013:**
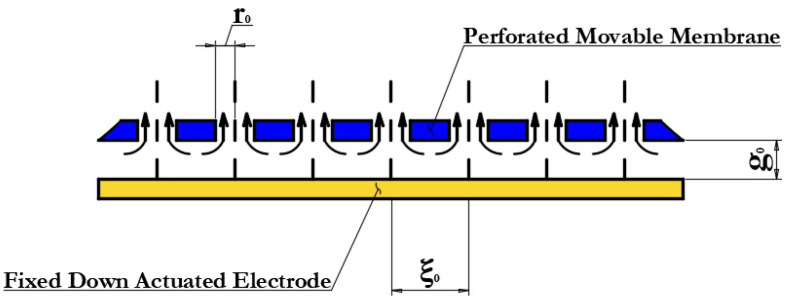
Schematic view of the air flow under movable membranes.

**Figure 14 micromachines-14-00477-f014:**
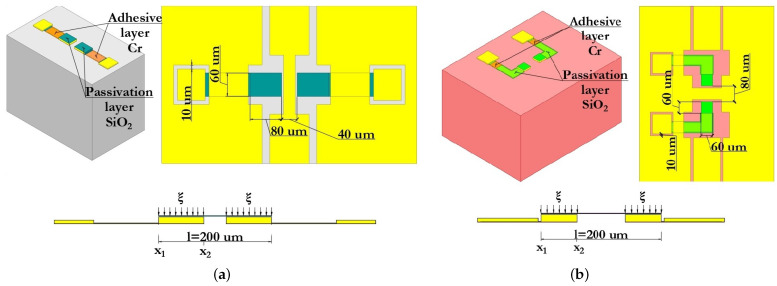
Schematic view of the structures of the fixed down actuation electrodes: (**a**) switch (A); (**b**) switch (B).

**Figure 15 micromachines-14-00477-f015:**
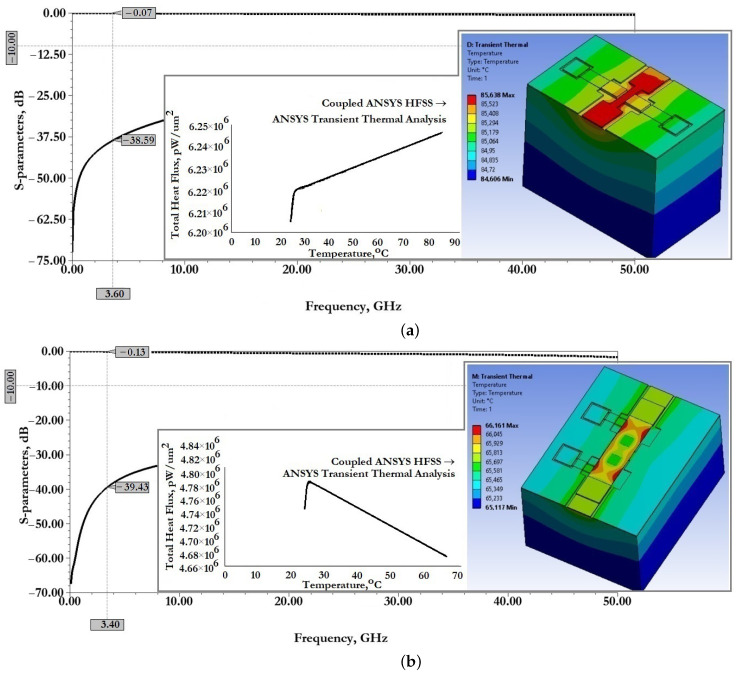
The results of EM and transient thermal modeling of CPW (A) and CPW (B) including fixed down actuation electrodes, contact pads and contact lines: (**a**) switch (A); (**b**) switch (B).

**Figure 16 micromachines-14-00477-f016:**
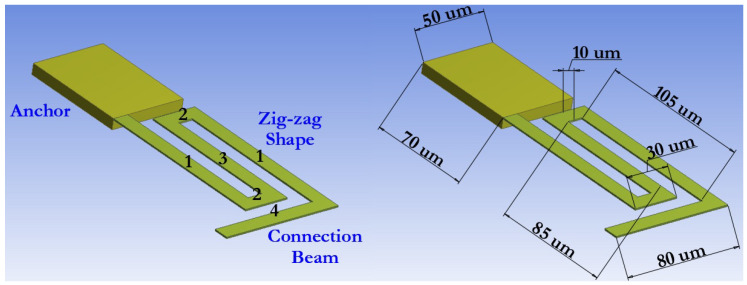
Schematic view of the developed zig-zag elastic suspension.

**Figure 17 micromachines-14-00477-f017:**
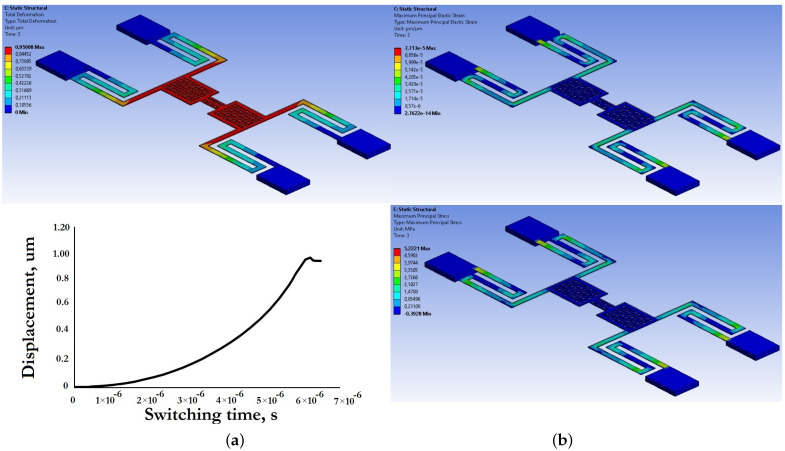
The results of modeling of electromechanical parameters of switch (A): (**a**) electrostatic displacement and switching time; (**b**) distribution of mechanical strain and mechanical stress.

**Figure 18 micromachines-14-00477-f018:**
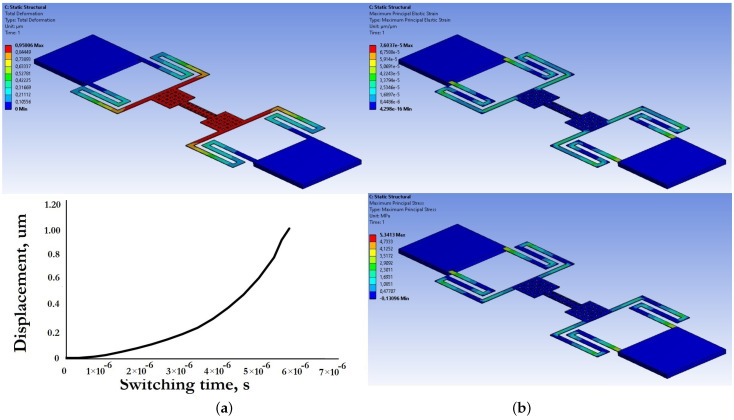
The results of modeling of electromechanical parameters of switch (B): (**a**) electrostatic displacement and switching time; (**b**) distribution of mechanical strain and mechanical stress.

**Figure 19 micromachines-14-00477-f019:**
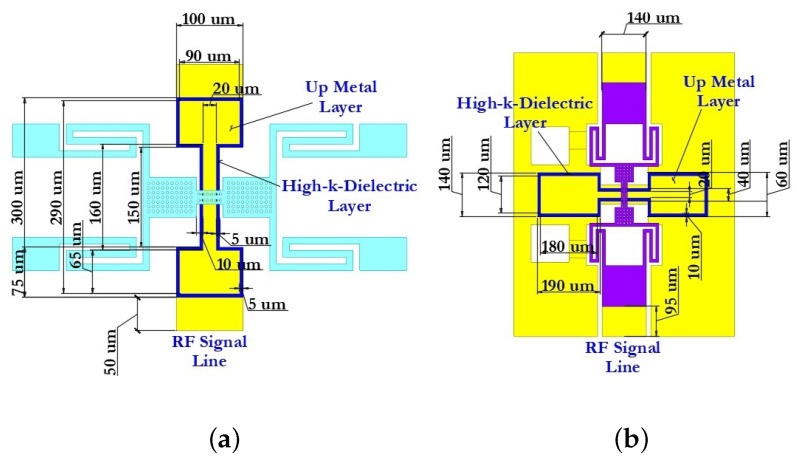
Schematic view of the developed additional fixed MIM capacitor: (**a**) switch (A); (**b**) switch (B).

**Figure 20 micromachines-14-00477-f020:**
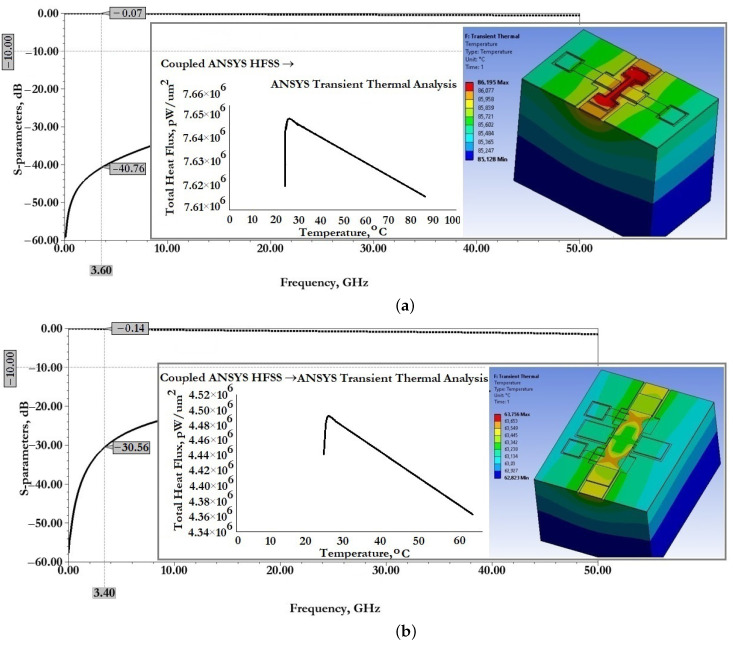
The results of EM and transient thermal modeling of CPW (A) and CPW (B) including fixed down actuation electrodes, contact pads, contact lines and MIM capacitor: (**a**) switch (A); (**b**) switch (B).

**Figure 21 micromachines-14-00477-f021:**
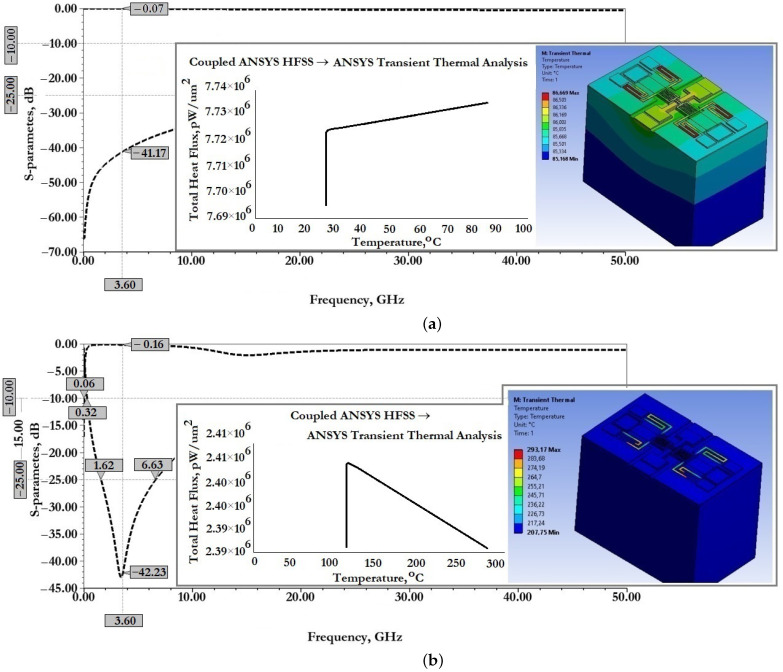
The results of EM and transient thermal modeling of the developed RF MEMS switch (A): (**a**) open state; (**b**) closed state.

**Figure 22 micromachines-14-00477-f022:**
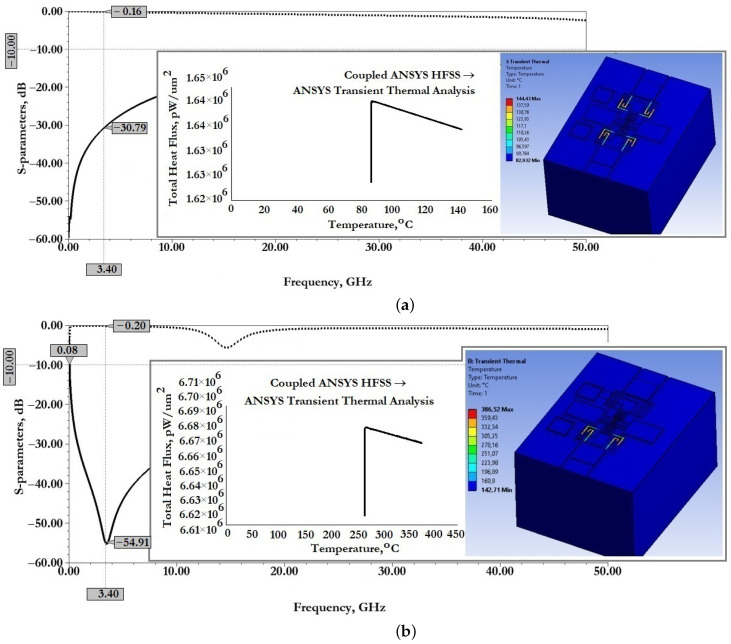
The results of EM and transient thermal modeling of the developed RF MEMS switch (B): (**a**) open state; (**b**) closed state.

**Figure 23 micromachines-14-00477-f023:**
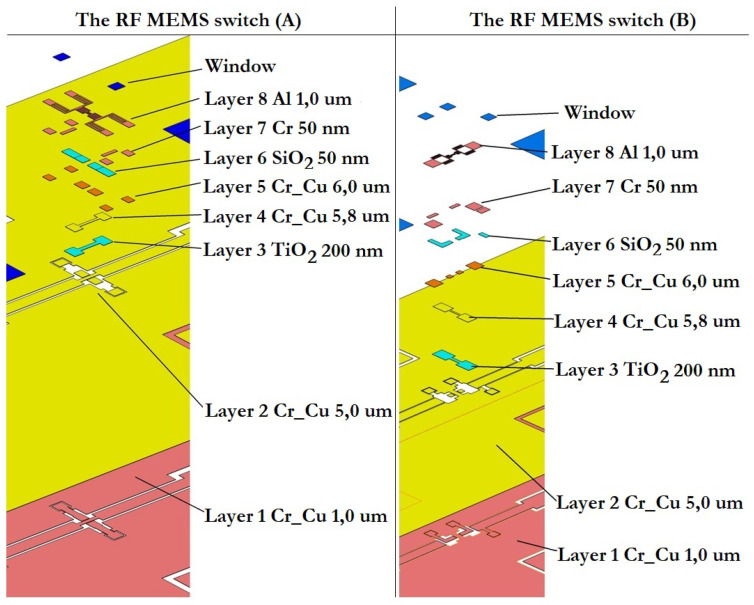
The technological layers in the manufacture of experimental samples.

**Figure 24 micromachines-14-00477-f024:**
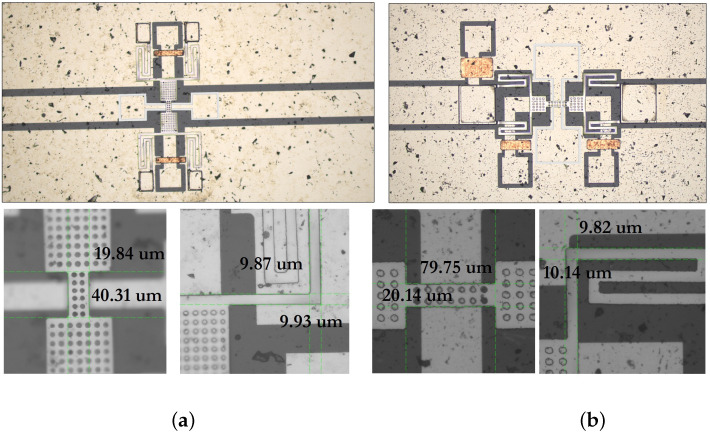
Photos of the manufacturing process of experimental samples: (**a**) RF MEMS switch (A); (**b**) RF MEMS switch (B).

**Figure 25 micromachines-14-00477-f025:**
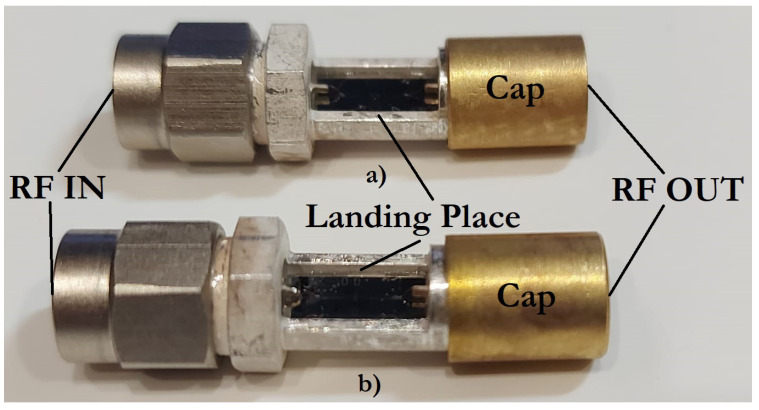
The manufactured experimental samples of RF MEMS switches: (**a**) RF MEMS switch (A); (**b**) RF MEMS switch (B).

**Figure 26 micromachines-14-00477-f026:**
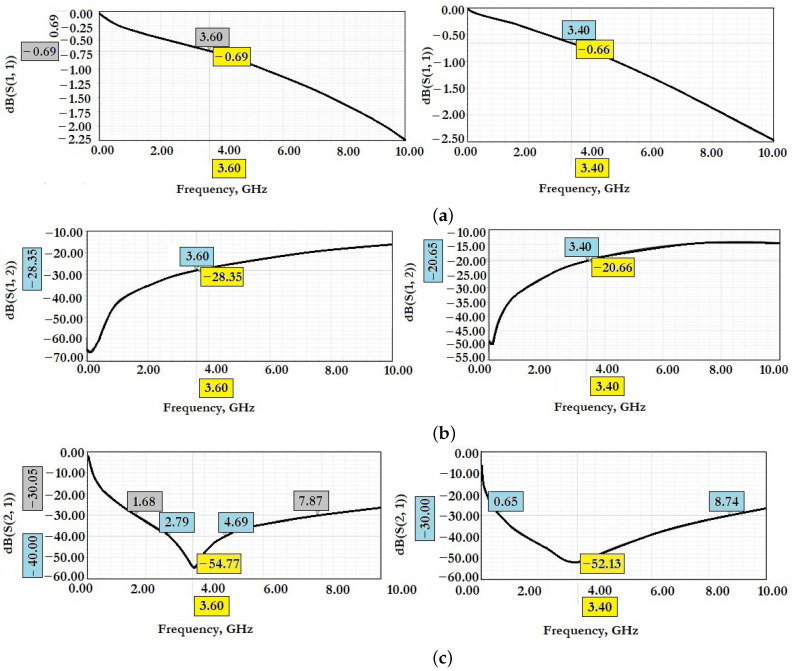
The results of the study of electromagnetic parameters of manufactured experimental samples of RF MEMS switch (A) and RF MEMS switch (B): (**a**) insertion loss in the open state; (**b**) reflection loss in the open state; (**c**) isolation in the closed state.

**Figure 27 micromachines-14-00477-f027:**
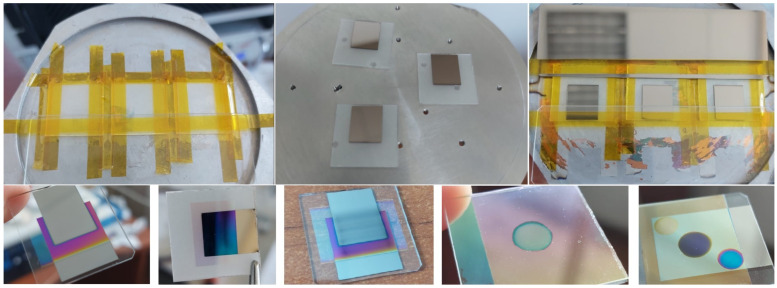
Photos of the process of applying capacitor structures in the experimental study of ${\mathrm{TiO}}_{2}$ thin films.

**Table 1 micromachines-14-00477-t001:** Properties of the most common materials for the CPW.

Material	Electrical	Thermal	Coefficient of
	Resistivity, ρ, Ω×m	Conductivity, *K*, W/m×K	Thermal Expansion, $\alpha_{T}$, $\left(10^{-6}{{(}^{\circ }C)}^{-1}\right)$
Aluminum	$2.90\;\times 10^{-8}$	222	23.6
Gold	$2.35\times 10^{-8}$	388	14.2
Copper	$1.72\times 10^{-8}$	315	17
Platinum	$10.60\times 10^{-8}$	71	9.1
Nickel	$9.50\times 10^{-8}$	70	13.3

**Table 2 micromachines-14-00477-t002:** The main parameters for choosing the material of the substrate.

Material	Dielectric	Electrical	Thermal	Coefficient of	Dielectric
	Constant	Resistivity	Conductivity	Thermal Expansion	Loss Tangent
	$\varepsilon_{d}$	ρ, Ω×m	K, W/m×K	$\alpha_{T}$, $\left(10^{-6}{{(}^{\circ }C)}^{-1}\right)$	$\tan\delta_{e}$
Quartz	3.8	$7\times 10^{17}$	13.8	5.5	≈$7.5\times 10^{-3}$
Glass	5–10	≈$10^{14}$	80	9	≈$2.5\times 10^{-3}$
GaAs	12.8	$10^{4}$–$10^{8}$	35–50	5.73	≈$4\times 10^{-3}$
${\mathrm{Al}}_{2}O_{3}$	11.3	$\approx 10^{13}$	50	4.7	$11\times 10^{-4}$
AlN	9.2	≈$10^{10}$	32.1	5.27	$2.1\times 10^{-3}$
BeO	6.76	≈$10^{11}$	33	8.9	$4\times 10^{-4}$
GaN	8.5	$7.8\times 10^{4}$	25.3	5.27	≈$10^{-2}$
InP	12.4	$10^{4}$	68	4.6	$3.2\times 10^{-2}$
LTCC	7.3	$10^{12}$	28	5.6	$7\times 10^{-4}$
SiC	9.6	≈$10^{6}$	20.7	11	$3\times 10^{-3}$
HRSi	11.7	$10^{7}$	24.7	9.2	≈$10^{-3}$

**Table 3 micromachines-14-00477-t003:** The main parameters for choosing the material of the movable mechanical structures.

Material	Young’s	Poisson’s	Electrical	Thermal	Coefficient of
	Modulus	Ratio	Resistivity	Conductivity	Thermal Expansion
	E,GPa	υ	ρ, Ω×m	K, W/m×K	$\alpha_{T}$, $\left(10^{-6}{{(}^{\circ }C)}^{-1}\right)$
Aluminum	69	0.33	$2.90\times 10^{-8}$	222	23.6
Gold	77	0.42	$2.35\times 10^{-8}$	388	14.2
Copper	115	0.33	$1.72\times 10^{-8}$	315	17
Platinum	171	0.39	$10.60\times 10^{-8}$	71	9.1
Nickel	204	0.31	$9.50\times 10^{-8}$	70	13.3
${\mathrm{Si}}_{3}N_{4}$	304	0.3	∼$10^{12}$	29	2.7
Mo	320	0.32	$5.20\times 10^{-8}$	142	4.9
${\mathrm{Al}}_{2}O_{3}$	380	0.22	∼$10^{13}$	39	7.4

**Table 4 micromachines-14-00477-t004:** The main parameters for choosing the material of the dielectric layer.

Material	Dielectric	Electrical	Thermal	Coefficient of	Young’s
	Constant	Resistivity	Conductivity	Thermal Expansion	Modulus
	$\varepsilon_{r}$	ρ, Ω×m	K, W/m×K	$\alpha_{T}$, $\left(10^{-6}{{(}^{\circ }C)}^{-1}\right)$	E, GPa
$\mathrm{Si}O_{2}$	3.8	$1\times 10^{14}$	1.4	5.6	71
${\mathrm{Si}}_{3}N_{4}$	7	$1\times 10^{14}$	3	9	304
${\mathrm{Al}}_{2}O_{3}$	10	$1.4\times 10^{14}$	39	7.4	380
AlN	9.14	$1.1\times 10^{14}$	16	7.7	330
$\mathrm{Hf}O_{2}$	25	$9\times 10^{13}$	1.1	6	57
${\mathrm{Ta}}_{2}O_{5}$	22	$1\times 10^{7}$	8	6.3	140
$\mathrm{Ti}O_{2}$	80	$1\times 10^{12}$	11.7	9	230
BST	800	$1\times 10^{5}$	12	9.4	1000
$\mathrm{Zr}O_{2}$	25	$1\times 10^{11}$	3.9	9.2	200

**Table 5 micromachines-14-00477-t005:** Specification of CPW (A) and CPW (B).

Component	Length, μm	Width, μm	Depth, μm	Material
CPW (A) $\left(G\;W\;G\right)$	20	100	20	Copper
Substrate	650	400	500	${\mathrm{Al}}_{2}O_{3}$
CPW (B) $\left(G\;W\;G\right)$	15	140	15	Copper
Substrate	900	700	500	${\mathrm{Al}}_{2}O_{3}$

**Table 6 micromachines-14-00477-t006:** Specification of the movable membranes of switch (A) and switch (B).

Component	Length, μm	Width, μm	Thickness, μm	Material
		Switch (A)		
Movable membrane	200	60	1	
Left part	80	60	1	Aluminum
Central part	40	20	1	
Right part	80	60	1	
		**Holes**		
**Form**	**Dimensions, μm**	$\Delta_{1},\mu m$	$\Delta_{2},\mu m$	**Numbers**
Circle	D=4	3	7	152
		Switch (B)		
Movable membrane	200	60	1	
Left part	60	60	1	Aluminum
Central part	80	20	1	
Right part	60	60	1	
		**Holes**		
**Form**	**Dimensions, μm**	$\Delta_{1},\mu m$	$\Delta_{2},\mu m$	**Numbers**
Square	4×4	6	8	68

**Table 7 micromachines-14-00477-t007:** Specification of the zig-zag elastic suspensions of switch (A) and switch (B).

Component	Length, μm	Width, μm	Thickness, μm	Material
		Switch (A)		
Zig-zagbeam	135	80	1	
$l_{b}$	105	10	1	
$l_{a}$	30	10	1	Aluminum
Connectionbeam	80	10	1	
		Numberof		
		meanders,n=1		
		$\mathrm{Air}\;\mathrm{gap},g_{0}=1$		
$K_{\mathrm{eff}},N/m$	$F_{d},\mu N$	$V_{p},V$	$t_{s},\mu s$	$t_{r},\mu s$
21.52	0.175	3.5	6.35	3.1
		Switch (B)		
Zig-zagbeam	135	80	1	
$l_{b}$	105	10	1	
$l_{a}$	30	10	1	Aluminum
Connectionbeam	80	10	1	
		Numberof		
		meanders,n=1		
		$\mathrm{Air}\;\mathrm{gap},g_{0}=1$		
$K_{\mathrm{eff}},N/m$	$F_{d},\mu N$	$V_{p},V$	$t_{s},\mu s$	$t_{r},\mu s$
19.35	0.15	5	6.5	3.2

**Table 8 micromachines-14-00477-t008:** The results of calculation of the MIM capacitors of switch (A) and switch (B).

$g_{0},\mu m$	δ	$\varepsilon_{r}$	$t_{d},\mu m$	$C_{\mathrm{MIM}},F$	$C_{\mathrm{MAM}},F$	$C_{r}$
Switch (A)						
1	37.25	80	0.2	$5.28\times 10^{-11}$	$3.54\times 10^{-27}$	14.901
Switch (B)						
1	116.5	80	0.2	$1.65\times 10^{-10}$	$3.54\times 10^{-27}$	46.601

**Table 9 micromachines-14-00477-t009:** Comparison of the results obtained.

Parameters	[59]	[60]	[61]	[62]	[63]	[64]	This Work
	**Lateral**	**Vertical**	**Lateral**	**Lateral**	**Vertical**	**Latching**	**Vertical**
$V_{p}$, V	57	75	15	32.6	26	38	3.5/5
Q	–	–	–	–	–	–	0.7
$d_{r}$	–	–	–	–	–	–	
$t_{s}$, μs	56	10	120	–	25	39.5	6.35/6.5
$t_{r}$, μs	40	5	150	–	13	94.8	3.1/3.2
$R_{c}$, Ω	1.5	1–2	–	–	∼1	–	0.7/0.62
$F_{\mathrm{range}}$	Sub	Sub	Sub	1–10	DC-30	DC-20	S-/
	6 GHz	6 GHz	6 GHz	GHz	GHz	GHz	C-, X-, Ku
$C_{r}$	–	–	–	–	–	–	14,901/46,601
**Open state**							
							**Theoretical**
$S_{11}$, dB	−0.28	−0.28	−0.31	−0.13	−0.18	−1.8	−0.07/−0.16
							**Experiment**
							−0.69/−0.66
(@GHz)	@6	@6	@6	@6	@2	@6	@3.6/@3.4
							**Theoretical**
$S_{12}$, dB	–	–	–	–	–	–	-41.17/-30.8
							**Experiment**
							−28.35/−20.66
(@GHz)							@3.6/@3.4
**Closed state**							
							**Theoretical**
$S_{21}$, dB	−38.4	−31	−36	−24.96	−38.8	−33.18	−42.2/−54.9
							**Experiment**
							−54.77/−52.13
(@GHz)	@6	@6	@6	@6	@2	@6	@3.6/@3.4
$P_{\mathrm{RF}}$, W	–	–	–	–	–	–	>1

## Data Availability

We did not report any data.

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
