# Peer review of "Investigation and Research of High-Performance RF MEMS Switches for Use in the 5G RF Front-End Modules"

_micromachines, 2023, doi:10.3390/mi14020477_

Round 1

Reviewer 1 Report

The Authors design a high-performance radio-frequency MEMS switch at 3.4-3.6 GHz. The results have been achieved by numerical simulations and experiments, demonstrating a good matching. The manuscript is very interesting and it deserves the publication. Here, my comments:

-          The first part of the manuscript is very interesting. It provides the main tool for a powerful MEMS switch design. As suggestion, the flow chart for the design could be published separately, as guidelines for engineers, and could reduce the number of pages of the revised manuscript. It’s a pleasure for me to support this potential “new” manuscript, that could include also the simulation approaches for MEMS devices.

-          About MEMS switch, the Authors should point out the main advantages/disadvantages with respect other technologies, as waveguide based (Design of a large bandwidth 2× 2 interferometric switching cell based on a sub-wavelength grating. Journal of Optics23(8), 085801, 2021), LiNbO3-based switches (Electro-optic X-switch using single-mode Ti:LiNbO3 channel waveguides Electron. Lett. 19 553–4, 1983), III–IV-based switches (Waveguided optical switch in InGaAs/InP using free-carrier plasma dispersion Electron. Lett. 20 228–9, 1984), semiconductor optical amplifier switching matrices (SOA gate array recirculating buffer for optical packet switching Proc. OSA Optical Fiber Communication Conf. 2008), liquid crystal-based switches (Variable-wavelength switchable Bragg gratings formed in polymer-dispersed liquid crystals Appl. Phys. Lett. 79 9–11, 2001), etc. Some of them such as liquid crystal, magneto-optic, bulk electrooptic and waveguide electro-optic have achieved good market success.

-          The Authors should better point out the used simulation tool.

-          The impedance and S-parameters graphs should be resized (see, i.e., Figure 10).

-          An extra peak at about 15 GHz arises in Fig. 21. The Authors should justify it.

-          A slight mismatch between theoretical and experimental results appears in Table 9 (open-state case). Please discuss on it.

Author Response

Dear Reviewer of MDPI Journal of Micromachines,

The team of authors of this manuscript and the staff of the Southern Federal University would like to thank you for taking the time to review and evaluate our manuscript "Investigation and research of high-performance RF MEMS switches for use in the 5G RF front-end modules" (micromachines-2155796). We also thank you for your objective assessment and comments, it always helps to raise the level and make the manuscript better. 

Thank you for your comment - the proposal to publish the proposed scheme - methodology for designing high-performance capacitive RF MEMS switches as a separate guide for engineers, but in this case we are limited to the proposal from the editorial board of the journal and would like to publish the entire manuscript "Theory" - "Approbation (Experiment)". Although if we consider the material for a separate manuscript or a guide for engineers on the proposed methodology, we could get no less voluminous manuscripts, because each of the proposed design stages deserves detailed consideration in theory and tools for modeling and calculation. At some design stages (for example, when calculating the parameters of isothermal and thermoelastic damping, dynamic parameters, parameters of the rebound of the movable membrane when the switch is closed or arbitrary latching during induced heating of the movable membrane, etc. d.) we have developed calculation programs in Matlab languages, which are not presented in the material of the manuscript for a significant reduction. This is the material of a separate manuscript, which we also plan to publish in this journal. 

We would also like to give brief answers to your following comments - remarks: 

- regarding the advantages and disadvantages of other technologies for the development of RF signal switching devices, for example, liquid crystal, magneto-optical, volumetric electro-optical and waveguide electro-optical, the comparison was not carried out and considers it completely corrective due to the fact that RF devices based on MEMS technologies (switches in one or more directions, switching matrices, variable inductors and variable capacitors, resonators) are the prospect of replacing traditional RF devices (in particular, systems and devices of 5G mobile radio-technology, both for mobile communication devices and for building a radio network architecture), made on the basis of semiconductor technology (PIN-diodes and FET-transistors), and can also be integrated as separate components into existing RF systems and devices today, without significant restructuring of these RF systems and devices, without which there is no do, for example, when using electro-optical, liquid crystal, magneto-optical devices. This concept of replacing traditional RF devices based on semiconductor technology and the integration of RF MEMS devices is proposed in the following works:

1) Nguyen C.T.-C. Microelectromechanical devices for wireless communications. Proceedings MEMS 98. IEEE. Eleventh Annual International Workshop on Micro Electro Mechanical Systems. An Investigation of Micro Structures, Sensors, Actuators, Machines and Systems, 1998, 1–7. 
2) Nguyen C.T.-C. Transceiver front-end architectures using vibrating micromechanical signal processors. Topical Meeting on Silicon Monolithic Integrated Circuits in RF Systems. Digest of Papers, 2001, 23–32. \\
3) Nguyen C.T.-C. RF MEMS for wireless applications. In 60th DRC. Conference Digest Device Research Conference, 2002, 9–12.
Nguyen C.T.-C. Integrated micromechanical circuits for RF front ends. Proceedings of the 32nd European Solid-State Circuits Conference, 2006, 7–16. \\
4) Nguyen C.T.-C. MEMS technology for timing and frequency control. In IEEE Transactions on Ultrasonics, Ferroelectrics, and Frequency Control , 2007, 54(2), 251–270.
5) Nguyen C.T.-C. MEMS-based RF channel selection for true software-defined cognitive radio and low-power sensor communications. In IEEE Communications Magazine , 2013, 51(4), 110–119.

- we cannot specify in detail the modeling tool at each of the stages of the proposed methodology for designing high-performance capacitive RF MEMS switches within one manuscript, since the volume of the manuscript is still limited and, as mentioned earlier, at some design stages (for example, when calculating the parameters of isothermal and thermoelastic damping, dynamic parameters, parameters of the rebound of a movable membrane during closure switch or arbitrary latching during induced heating of the movable membrane, etc. d.) we have developed calculation programs in Matlab languages, which are not presented in the material of the article for a significant reduction. However, the tools for calculation and modeling at each of the design stages of the presented methodology are Matlab and the ANSYS Multiphysics subsystems (HFSS, Maxwell, Structural, Modal, Transient Thermal);

- all the graphs of scattering parameters (S-parameters) presented in the article have been corrected to eliminate any inconsistencies in the axis signatures on the graph, and the font size of all the signatures on the graphs has been increased so that all the graphs presented are clear and not confusing. The graph of S-parameters shown in Figure 10 has also been edited;

- a certain peak at the RF signal frequency of about 15 GHz according to the results of the simulation presented in Figure 21, as the practice of finite element modeling shows, arises due to the influence of the inductance of the elastic suspension elements of the movable membrane. By reducing this value of inductance (reducing the number or geometric dimensions) of meanders, it is possible to achieve less influence of this inductance on the resonant frequency in any frequency band, but in this case it will lead to an increase in the value of the control voltage. But the presented RF MEMS switch is designed to operate at a central resonant frequency of 3.6 GHz and the S-frequency range, and no additional parasitic influence is observed at this frequency and frequency range;

- some incorrect data presented in Table 9 have been corrected, but as for some discrepancy between the theoretical and measured experimental data regarding electromagnetic parameters, everything is explained by the fact that a specialized metal package for RF applications (up to 10 GHz) with coaxial terminals was used to study the manufactured experimental samples of RF MEMS switches, which facilitates the process measurements of electromagnetic parameters. However, in order to install the manufactured experimental samples of RF MEMS switches in this package, we needed to mount them on a special adapter board placed in this package, and then solder the corresponding contacts with a thin gold wire. Unfortunately, at the moment we do not have the opportunity to measure the electromagnetic parameters of RF MEMS switches or RF MEMS structures without using a specialized RF package (at the crystal- or wafer-level), which would be more reliable with theoretical data.

We hope that our brief explanation has clarified your comments - remarks.

Sincerely, the scientific team of the authors of the manuscript of the Southern Federal University.

Alexey Tkachenko <alexeytkachenko@sfedu.ru>
Igor Lysenko <ielysenko@sfedu.ru>
Andrey Kovalev <avkovalev@sfedu.ru>

Author Response

Dear Reviewer of MDPI Journal of Micromachines,

The team of authors of this manuscript and the staff of the Southern Federal University would like to thank you for taking the time to review and evaluate our manuscript "Investigation and research of high-performance RF MEMS switches for use in the 5G RF front-end modules" (micromachines-2155796). We also thank you for your objective assessment and comments, it always helps to raise the level and make the manuscript better.

We would also like to give brief answers to your following comments - remarks:

1) The "Abstract" section has been completely redesigned and described in detail. A brief description of the designs of high-performance RF MEMS switches designed using the proposed design methodology and manufactured experimental samples, the results of their experimental study, has been added.

2) Figures 11, 15, 20, 21, 22 have been revised. The font of signatures and designations has been enlarged to increase their clarity and visibility.

3) Graphic information has been added in Figure 7 on the geometric dimensions of the designed and then manufactured experimental samples of RF MEMS switches at the level of a single crystal.

4) Demonstration and methodology of selecting the appropriate material for a certain part of the design of the projected RF MEMS switches allows you to achieve an advantage - reducing or increasing a certain key parameter (parameters). Table 9 shows a comparison of designs of high-performance RF MEMS switches designed using the proposed design methodology and manufactured experimental samples for use in 5G NR1 mobile radio communication devices and systems with designs of RF MEMS switches (designed or manufactured) for the same application and frequency range.

5) Figure 2 is a methodology for designing high-performance capacitive RF MEMS switches - a kind of design route that can be used as a separate guide for engineers, but in this case we are limited to a proposal from the editorial board of the journal and would like to publish the entire manuscript "Theory" - "Approbation (Experiment)". Although if we consider the material for a separate manuscript or a guide for engineers on the proposed methodology, we could get no less voluminous manuscripts, because each of the proposed design stages deserves detailed consideration in theory and tools for modeling and calculation. At some design stages (for example, when calculating the parameters of isothermal and thermoelastic damping, dynamic parameters, parameters of the rebound of the movable membrane when the switch is closed or arbitrary latching during induced heating of the movable membrane, etc. We have developed calculation programs in Matlab languages, which are not presented in the material of the manuscript for a significant reduction, but are consistently and briefly covered in the submitted manuscript. This is the material of a separate manuscript, which we also plan to publish in this journal.

6) In Figure 23 information on the layer-by-layer decomposition of manufactured experimental samples of designed RF MEMS switches is edited, indicating the material, the material of the adhesive sublayer and indicating the thickness of each of the layers.

7) A fragment has been added to Figure 4 a), which schematically shows a one-dimensional model of the displacement of the movable membrane during electrostatic activation, and also in this section 3.1. a text description of the process of electrostatic activation has been added (line 502-519). Also, the behavior of a movable membrane under the action of electrostatic activation is demonstrated in Figure 17 and Figure 18 using finite element analysis in the ANSYS Multiphysics (Structural - Transient Structural Analysis) software environment. 

8) The data presented in Figure 26 are experimental - measurements of electromagnetic parameters of manufactured experimental samples of RF MEMS switches. Figure 26 has been redesigned: the font of signatures and designations has been enlarged to increase their clarity and visibility, the structure of the figure has been changed. A fragment has been added to Figure 26, which graphically shows the distribution of reflection losses in the open position of the manufactured experimental samples of RF MEMS switches (Figure 26 b)). The discrepancy between the experimental data presented in Figure 26 and those shown in Table 9 has also been edited.

We hope that our brief explanation has clarified your comments - remarks.

Sincerely, the scientific team of the authors of the manuscript of the Southern Federal University.

Alexey Tkachenko <alexeytkachenko@sfedu.ru>
Igor Lysenko <ielysenko@sfedu.ru>
Andrey Kovalev <avkovalev@sfedu.ru>

Round 2

Reviewer 2 Report

1.      The abstract should include a short description of the background, methods, results, and conclusions. The authors have expanded their abstract with them, the updated abstract with around 500 words, while the journal required “The abstract should be a total of about 200 words maximum”.

2.      From Figure 26(a), S11 in the open-state are -28.35 dB @ 3.6 GHz and -20.66 @ 3.4 GHz. From Figure 26 (b), S12 in the open-state are -0.69 @ 3.6 GHz and -0.66 @ 3.4GHz. But in Table 9, S11 is -0.7/-0.7 dB @ -3.4/3.6 GHz, and S21 is -28.35/ -20.7 dB @ 3.4/3.6 GHz. There are some mistakes.

3.      Although there are several figures about the schematic or topology of the proposed switches, Figures 4, 5, 6, 7, 14, 19, and 23, I am still confused about the structure. Figure 23 shows the structure for each layer of the structure, but it's hard to imagine the whole structure of the switches.

Author Response

Dear Reviewer of MDPI Journal of Micromachines,

The team of authors of this manuscript and the staff of the Southern Federal University would like to thank you for taking the time to review and evaluate our manuscript "Investigation and research of high-performance RF MEMS switches for use in the 5G RF front-end modules" (micromachines-2155796). We also thank you for your objective assessment and comments, it always helps to raise the level and make the manuscript better.

We would also like to give brief answers to your following comments: 

- the content of the "Abstract" section has been revised. A very brief description of the manuscript, methods and results is presented. The content of this section has been reduced to the volume that the journal regulates (no more than 200 words);

- the erroneous discrepancy between the experimental data presented in Figure 26 and Table 9 has been corrected. Figure 26 has also been edited, namely part a) and part b) - the designations on the axes and the caption to the figure have been edited. In addition, errors in the conclusions after Figure 26 have been edited, namely lines 905-919;

- in the section "Conclusion" in Table 9, erroneous data regarding the experimental data in Figure 26, as well as in the text in lines 959-965 are edited.

Some explanation about the structure of the developed RF MEMS switches:

- The manuscript in Figure 5 (a) and Figure 5 (b) presents an isometric three-dimensional representation of the developed RF MEMS switch designs. The geometric dimensions of these structures - i.e. individual crystals (substrates) are shown in Figure 7 and Table 5. The dimensions of individual structural elements, such as a movable metal membrane, elastic suspension elements, fixed lower control electrodes, etc. are shown in Figure 12, 14, 16, 19 and in Table 6, 7, 8. Figure 23 shows a three-dimensional decomposition of the developed RF MEMS switch designs by layers, indicating the thicknesses and heights of each of the layers.

We declare that this manuscript is original, has not been previously published and is not currently being considered for publication elsewhere, and we also hope that you will find our manuscript suitable for publication after making minor changes.

Sincerely, the scientific team of the authors of the manuscript of the
Southern Federal University.

Alexey Tkachenko <alexeytkachenko@sfedu.ru>,
Igor Lysenko <ielysenko@sfedu.ru>,
Andrey Kovalev <avkovalev@sfedu.ru>.
